

# Application of an improved global-scale groundwater model for water table estimation across New Zealand.

Rogier Westerhoff[1,2], Paul White[1], and Gonzalo Miguez-Macho[3]

[1]GNS Science, Taupo, New Zealand
[2]Deltares, Utrecht, The Netherlands
[3]University of Santiago de Compostela, Spain

**Correspondence:** R.S. Westerhoff (r.westerhoff@gns.cri.nz)



**Abstract.** Many studies underline the importance of groundwater assessment at the larger, i.e., global, scale. The large-scale models used for these assessments are often simplified and typically not used for smaller-scale, i.e., catchment-scale, studies, because hydrology and water policy are traditionally best constrained at the catchment scale, and because large-scale models are too uncertain for that scale. However, smaller-scale groundwater models can still have considerable uncertainty, especially

in data-sparse areas. There is a potential for larger-scale models to constrain the uncertainty for small-scale models. That is because they can provide an extra source of information in data-sparse areas, such as the initial estimate of hydraulic head. Large-scale models, often quick and simple, can thus take away some of the computational burden of local and more sophisticated applications. The problem of this approach is that model uncertainty of large-scale models is often too large, because the quality of their, coarse and global-scale, input data is large, and often inconsistent with the input data of local

models. What is needed is an approach where large-scale and local models can meet 'in the middle'.

This study uses an existing, global-scale, groundwater flow model. It feeds that model with national input data of New Zealand terrain, geology, and recharge. It then builds the first New-Zealand national-scale groundwater model. The resulting nationwide maps of hydraulic head and water table depths show that the model points out the main alluvial aquifers with fine spatial detail (200m grid resolution). The national input data and finer spatial detail result in better and more realistic variations

of water table depth than the original, global-scale, model outputs. In two regional case studies in New Zealand, the hydraulic head matches the available groundwater level data well. The nationwide water tables show that the model is mostly driven by the elevation (gravity) and impeded by the geology (permeability).

The use of this first New Zealand-wide model can aid in provision of water table estimates in data-sparse regions. The national model can also be used to solve inconsistency of models in areas of trans-boundary aquifers, i.e., aquifers that cover

more than one region in New Zealand. Shortcomings of the model are caused by the simplified model properties, but also by the accuracy of input data. Future research should therefore not only focus on further improvements of model equations, but also improved estimation of hydraulic conductivity and the digital elevation model, especially in areas of shallow groundwater level. We further surmise that the findings of this study, i.e., application of a global-scale models at smaller-scales, will lead to subsequent improvement of the global-scale model equations.



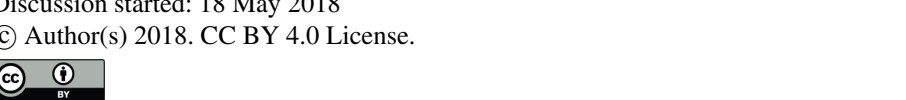



*Copyright statement.* TEXT



# 1 Introduction

Groundwater is a key water resource to many countries in the world, providing water for irrigation, domestic consumption and industry. Groundwater also provides baseflow to streams and rivers, which, in drier times, helps sustain ecology when rainfall runoff is low. Many studies underline the importance of global-scale assessment of groundwater (Wada et al., 2012), its role in a changing climate (Taylor et al., 2012; Green et al., 2011), and its current depletion (Gleeson et al., 2012; Richey et al., 2015).

Groundwater models used for global-scale assessments are simplified. These models generally use coarse input data or embed a simplified model algorithm. For example, de Graaf et al. (2015, 2017) apply a model with approximately 10 km resolution and global-scale input data and Fan et al. (2013a) apply a global model of water table estimates with grid cells of approximately 1 km, using simplified flow equations and subsurface parametrisation.

Most practical examples of groundwater studies are at the catchment scale, where hydrology is traditionally best constrained (because inflows and outflows can be best estimated at that scale). Water policy also typically fits the catchment scale and, in most countries, decision making is based on hydrological studies of this scale.

Despite the abundance of groundwater models at the catchment scale (e.g., Gusyev et al., 2013; Wang et al., 2008; Oude Essink et al., 2010), the uncertainty of those models is high and therefore groundwater is still one of the least understood resources ("the hidden part of the water cycle", Margat and van der Gun, 2013). Many studies have added to a better definition of the uncertainty of catchment-scale groundwater flow models (e.g., Delhomme, 1979; Moore and Doherty, 2005; Rojas et al., 2008; Knowling and Werner, 2016). Groundwater models are less reliable in data-sparse regions, and uncertainty of groundwater models is largely dependent on the availability of data, such as water levels in wells, or geophysical measurements (e.g., Van Overmeeren, 1998; Loke and Barker, 1996; Siemon et al., 2009; Danielsen et al., 2003).

National-scale groundwater information in New Zealand is important because guidelines on groundwater allocation are defined by the national government (Ministry for the Environment, 2013, 2008), thus requiring a nationwide overview. Furthermore, national-scale hydrology models in New Zealand (Topnet: Bandaragoda et al., 2004) require groundwater information to calculate the interaction of groundwater with surface water, such as baseflow or flow loss from rivers.

However, obtaining a national overview of New Zealand's groundwater resources still faces many challenges. In New Zealand, water resources and allocation are managed by regional councils, and regions mostly have the same boundaries as the catchments. Regions do not have agreed and consistent modelling approaches, data formats, and data availability (Westerhoff et al., 2018), causing 'trans-boundary' issues, such as drinking water source protection and catchment boundary definition. Where some regions are relative data-rich, others do not have enough information to obtain a detailed groundwater flow model. Development of individual advanced groundwater flow models for each region, compatible across regions, is recommended but would take many years to complete in the current policy framework, with significant efforts required in aligning regional and national policy makers. Also, the combined model run time of those advanced, and therefore computationally demanding, models would be a large computational burden.

This study describes the development of a groundwater model at the national scale of New Zealand. Our model is inspired by the global-scale Equilibrium Water Table (EWT) model (e.g., Fan et al., 2013a), with improved input data (i.e., national





input datasets of elevation, geology, recharge) and adjusted resolution and computational efficiency. It is therefore called the National Water Table (NWT) model. A review of the EWT model is first given, after which the improved NWT model method is explained. Results simulated at the national scale are presented, and furthermore evaluated at the catchment scale by comparing simulated groundwater depths to observations in two regional case studies. In addition, this paper discusses the relevance of the

5  NWT model to solve inconsistency between regional models, and the strength and weakness of the simplifying assumptions in the model. We also discuss the importance of model input components (e.g., terrain model and hydraulic conductivity).



## 2   Review of the EWT model

The Equilibrium Water Table (EWT) model is a steady-state groundwater model that calculates water table depth (in metres below ground level: mBGL) and water table elevation (in metres above sea level: masl) at the global scale using a variety of ground-based, satellite-observed and modelled parameters (Fan et al., 2013a, 2007; Miguez-Macho et al., 2007, 2008; Fan

and Miguez-Macho, 2010b). The water table ("the level in the saturated zone at which the pressure is equal to the atmospheric pressure", Heath, 1995) represents a long-term average at a broad scale without human-induced effects, e.g., pumping, draining and irrigation. The model calculates a single water table, and therefore it generally cannot represent smaller-scale water tables and piezometric surfaces in perched or multi-layered aquifers.

To calculate the water table, the long-term balance between the groundwater recharge and horizontal groundwater flow is

calculated using a simple groundwater flow equation. The groundwater flow is constrained by the sea level, assuming that the hydrology is in equilibrium with the climate and sea level (Marshall and Clarke, 1999). Model input data include a global topography model, a long-term time series of global groundwater recharge, and a global soil model. Ground-observed ground-water level data have been used for calibration of hydraulic conductivity on a continental scale (Fan et al., 2013b). However, no New Zealand data were used in that calibration. Details of the method are described in Fan et al. (2013b) and summarised

in appendix A.

Westerhoff and White (2013) evaluated water table depths from the EWT model in New Zealand. Generally, they found that the model correctly estimated shallow water tables in New Zealand's alluvial aquifer systems, such as the Hauraki Plains, Heretaunga Plains, and the Canterbury Plains (Fig. 1). However, they also found several issues in the Canterbury Plains, which included a bias to calculation of shallow water table depth, i.e, EWT depths were too shallow compared to ground

observations at many locations. They also described the importance of better terrain models, as the uncertainty of the global-scale datasets is generally 10 m or more (Gesch et al., 1999; Harding et al., 1999; Rodríguez et al., 2006). Therefore, Westerhoff and White (2013) recommended improvement in the method and its input data, which included better model convergence criteria; representation of terrain; estimation of hydraulic conductivity, i.e., a better representation of the underlying geology; rainfall recharge data that is relevant to the New Zealand climate.



## 3  Methodology

The National Water Table (NWT) model is based on the EWT model, but uses adjustments as proposed by other global-scale groundwater studies (Gleeson et al., 2011; de Graaf et al., 2015) and uses national-scale New Zealand input data. Our model improvements include: model resolution; a national terrain model; national recharge estimates; and hydraulic conductivity
estimates based on the geological map of New Zealand.

### 3.1  Model resolution, initial conditions and convergence criteria

The model grid cell size was chosen as 200 m, in the New Zealand Transverse Mercator (NZTM) coordinate system. All input data were gridded at this cell size, either by averaging (if the cell size of the original input data was smaller than the model grid cell size) or by its nearest value (if cell size of the original input data was larger than the model grid cell size). The model
was run in daily time steps. Initial estimates of water table depth depth and elevation for the NWT model were set equal to the EWT water table depth and elevation, i.e., the results of the EWT model.

The EWT method had a convergence criterion, where the iterative calculation stops when the water table reaches an equilibrium, i.e., when lateral groundwater flow equals recharge for every model cell. This criterion has been changed in the NWT method; the model currently runs for 100 years. Appendix B explains that convergence tests showed that running for more
than 100 years improves model convergence, but does not necessarily lead to better results, because most of the changes with increased runtime occur in shallow water features, which in reality are mostly drained by humans. Additionally, feeding in recharge that contains uncertainty (Westerhoff et al., 2018) causes the equilibrium to probably never be reached.

### 3.2  National terrain model

The Geographx New Zealand DEM 2.1 (Geographx, 2012) is a national digital elevation model with an 8 m grid resolution,
which combines New Zealand-based topographic data and satellite data from the Shuttle Radar Topography Mission (SRTM: USGS, 2006). More accurate elevation data are available at the regional scale, but the Geographx model is the best national terrain model.

### 3.3  National rainfall recharge

Rainfall recharge to groundwater was taken from a national monthly New Zealand rainfall recharge dataset (Westerhoff et al.,
2018). For the purpose of feeding recharge into the model with daily time steps, these values were converted to mean daily recharge (Fig. 3).

### 3.4  Hydraulic conductivity

The NWT model uses data from the national geological map of New Zealand. The steps to derive saturated hydraulic conductivity K values suitable for NWT model input are: estimation of near-surface K; (optional) local calibration of near-surface K;
and estimation of K over depth. These steps are described below.





### 3.4.1 Estimation of near-surface K

The NWT method applies geology data, i.e., the deeper subsurface underlying the soil, from the national 1:250,000 geological map of New Zealand (QMAP: GNS Science, 2012) to calculate near-surface K. The QMAP is a GIS-based digital map that shows 'surface geology', i.e., the geology of the subsurface up to approximately 10 metres depth. The polygon attributes of main rock type, secondary rock type and age were used to estimate K (Westerhoff et al., 2018) and were partly based on a classification method described by Gleeson et al. (2011), who estimated hydraulic permeability $\kappa$ [m$^2$] and its standard deviation for a range of hydrolithological classes. This method was put in the New Zealand context by Tschritter et al. (2016), who defined 10 hydrolithological classes and their associated $\kappa$ (Table 1). All QMAP rock type attributes were summarised to 183 'dictionary' values, which were then assigned to permeabilities with a look-up table. Permeabilities of main rock type and secondary rock type were averaged, with main rock type weighted twice as much as secondary rock type.

Saturated hydraulic conductivity $K$ [m/day] was calculated following Freeze and Cherry (1979) (their Eq. 2.28):

$$K = 86400 \frac{\kappa \rho g}{\mu} \tag{1}$$

where:

- $\mu$ is the dynamic viscosity of freshwater at 13°C (=1.2155×10$^3$ kg /m s);

- $\rho$ is the density of fresh water (=1000 kg / m$^3$);

- $g$ is the gravitational constant (=9.90 m$^2$/s);

### 3.4.2 Local calibration of near-surface K

Values of the original K (Eq. 1) were adjusted (to K$_{adj}$) in iterative calibration steps in areas where water table depth and elevation were deemed to be sufficiently known from local measurements:

$$K_{adj} = \zeta \left( h_{insitu} - h_{model} \right); \quad |K_{adj} - K| < 100 \tag{2}$$

The calibration damping factor $\zeta$ was chosen by trial and error to be 0.66/day. The maximum adjustment allowed per calibration time step was 100 m/day. This calibration was run every 10 years of the 100-year model run time. This procedure was only used in the case where ample ground-observed water level data were available for this study (Canterbury). Before calibration, the ground-observed water level point data were gridded to an interpolated surface within the area containing ground observations.

### 3.4.3 Estimation of K over depth

Near-surface K was assumed to be represented by the QMAP-derived near-surface K (Fig. 2). Deeper than 10 m, an exponential decrease of hydraulic conductivity over depth was assumed, similar to Eq. A1. As cell resolution of the NWT model is 200 m, the values of $a$, $b$ and $f_{min}$ of Eq. A2 were changed accordingly, to 75, 150 and 4, respectively. These values were also used by a 200 m resolution EWT model in the Amazon basin (Fan and Miguez-Macho, 2010a) .



## 4   Results

### 4.1   National water table maps

National maps of water table depth (Fig. 4) and water table elevation (Fig. 5) show that water table depth is relatively deep in higher mountainous regions, but water table elevation clearly follows the terrain elevation. The NWT water table depths show

the locations of the main alluvial aquifers, similar to the EWT model. The improved NWT water table depth clearly follows the aquifer delineations of White (2001), e.g., in the Canterbury Plains and Heretaunga Plains (Fig. 6), and also are detailed enough to possibly improve this delineation in some areas. The finer detail in the NWT model also results in a more realistic local variation of water table, compared to the EWT model results (Fig. 1). Many very small shallow water table features are found in between areas where water tables are deeper. These features resemble stream valleys, as the more detailed terrain

model allows the model to discharge more water to the surface in these valleys. The finer spatial detail is best appreciated when zooming in to smaller regions. Therefore, evaluation analyses are described for two regional case studies (Canterbury Region, section 4.2 and the Waipa River catchment, section 4.3).

### 4.2   Evaluation of NWT water table in the Canterbury Region

The Canterbury Plains is New Zealand's largest alluvial aquifer system. Nationwide, regional groundwater allocation is largest

in the Canterbury Region (Rajanayaka et al., 2010), with the majority of groundwater use in the Plains. Surface geology in the Canterbury Region is dominated by metamorphic rock and floodplains (Fig. 7). Metamorphic rock includes a large area of greywacke that form much of the Southern Alps. Aquifers in the higher Plains are mostly unconfined; in the lower Plains they are typically confined and supply groundwater to extensive networks of spring-fed streams around Lake Ellesmere and Christchurch City. These confining conditions are due, in part, to the occurrence of marine sediments deposited near the current

coast during interglacial periods. However, the pattern of groundwater flow is not influenced by any extensive, low-permeability layers of sediment within the upper 100 to 150 m below the water table (Hanson and Abraham, 2009). Therefore, application of the NWT (and EWT) methods, which assume unconfined aquifers, is justified in the Canterbury Plains.

Groundwater level observations from 8664 time series in Canterbury wells, acquired from 1894 to 2013 (Palmer, 2013), have been used in this research. Most of these wells are located in the Canterbury Plains. It is assumed that these observations

represent the water table and can thus be compared to NWT water table depth and water table elevation. All groundwater level observations were corrected for the measurement reference level and quality checked. The following data were rejected: missing or 'no data' values; time series that ended before 1980; time series with durations shorter than one full year; and data containing less than three measurements (Westerhoff and White, 2013). The result is a dataset containing groundwater depth and elevation from 3286 wells. NWT water table depth and elevation covering the locations of these wells were sampled from

both the NWT and EWT datasets. Median observed groundwater level for all wells were calculated. From here, these will be referred to as 'ground-observed water level'. The national Geographx elevation model was used for conversion of groundwater depth to groundwater elevation.





NWT water table depths are similar to ground-observed water depth: for example, 24%, and 53% of NWT water table depths are within 1 m and 3 m from ground-observed water depth, respectively (Table 2). This is an improvement to the EWT water table depths, where this was 13%, and 42%, respectively. Also, the NWT model shows significantly less large discrepancies than the EWT model. For example, 3.4%, 0.18% and 0.03% of the NWT water table depths differed more than 50, 100 and

150 m, respectively, from the ground-observed water depth. For the EWT water table depths, these statistics were 6%, 1.2% and 0.05%.

Correlation of NWT water table elevation with ground-observed water level (masl) is high (R=0.99, Fig. 8, right), but does show local differences, which are shown more clearly in the crossplot of water table depth (Fig. 8, left). Westerhoff and White (2013) showed that evaluation of the EWT model revealed similar discrepancies in the same area (i.e., the area between the

Ashburton and Rakaia rivers, Fig. 7). Although the discrepancies are smaller for the NWT model than for the EWT model in these areas (Fig. 9), they occur at the same locations. Possible explanations for these discrepancies are considered in the discussion.

## 4.3   Evaluation of NWT water table in the Waipa River catchment

The Waipa River is a tributary of the Waikato River, in the Waikato Region, New Zealand (Figure 10). Increasing agricultural

land use and deforestation, mainly in the low-lying Hamilton Basin, could potentially result in the deterioration of water quality in the Waipa River catchment, according to Rawlinson (2014), who performed a review on existing information in the catchment. They calculated a map of water table elevation using observed groundwater elevation from 758 wells located in the catchment. However, these observations are sparse, because: temporally, data mostly consists of few measurements; and spatially, data are concentrated more in some areas than others, i.e., most wells are located in the Hamilton Basin. The NWT

model can thus provide a better spatial insight of the water table in the catchment, provided that they correlate well enough with the ground observations.

Two NWT model runs were evaluated. The first run included the national terrain model, while the second run used a terrain model based on LiDAR data, which was resampled to 100 m grid resolution. Correlations of the first run with ground-observed data were R=0.25 and R=0.95 for water table depth and water table elevation, respectively. For the second run, they were

R=0.41 and R=0.97, respectively (Fig. 11). However, model run time increased by a factor of six due to the higher resolution terrain model. Water table depth shows significantly lower correlation than water table elevation (as was also the case in the evaluation in the Canterbury Region). The discussion of this paper describes possible causes for these differences.

NWT water table depths clearly demonstrate the location of shallow water tables in the low-lying basin area (Fig. 12). The spatial pattern of NWT water table elevation is a good visual match to the interpolated ground-observed water table elevation

well (Fig. 13). In addition, the NWT map shows more detail than the interpolated ground-observed surface and calculates water table depth at many places where there are no ground observations.





## 5 Discussion

The NWT model estimates water table depth and water table elevation with 200 m grid resolution at the national scale. Model equations are based on the global-scale EWT model, with adjusted model parametrisation and national input datasets. Because of these improvements, NWT-derived water table depths show the areas of alluvial aquifers, including their variations and their

groundwater discharge to surface, with higher spatial resolution than the EWT model. The NWT water table elevation shows excellent correlation with ground-observed water level data. The NWT model is currently the only nationwide groundwater model in New Zealand and it is able to estimate the water table at places where there are no ground observations. In addition, it shows more detail than most other interpolated water table surfaces.

The NWT model includes all catchments of the mainland of New Zealand. Because water is primarily regulated at the re-

gional level, regional models can show different results (e.g., groundwater catchment delineation) at regional boundaries. These inconsistencies can lead to trans-boundary issues, such as catchment boundary definition and source protection. A fundamental role of groundwater science is to identify and characterise groundwater catchments of water supplies. Better delineation of groundwater catchments is therefore essential for source protection. Source protection is clearly an important issue associated with the prevention of water-borne diseases and a national environmental standard for source protection, including groundwa-

ter, has been proposed for New Zealand Ministry for the Environment (2009, 2016). Where two regional groundwater models show different results, the NWT model can be used as a 'third' model to help solve those inconsistencies. This makes the NWT model relevant to solve trans-boundary issues and to assist in provision of nationally consistent groundwater data.

The advantage of the simplified NWT model (compared to catchment-scale groundwater flow models) is that computation of the water table across all catchments is relatively fast. The NWT model can therefore provide useful preliminary water table

information for catchment-scale numerical groundwater flow models, in both data-sparse and data-rich regions. Furthermore, the NWT model inputs can provide other initial estimates that are useful to other models. For example, data of hydraulic conductivity and recharge are also nationwide.

The disadvantage of simplifying assumptions in the NWT model is that more complex groundwater features are not handled well. Currently, the model does not include confining layers reliably, and neither does it incorporate fractures and groundwater

age. This is mostly due to the simplifying assumption of K over depth. The assumption of decrease of K with depth is based on unconfined water movement. Water table depth does therefore not necessarily equal groundwater level in confined aquifers, as the confining layer holds the water under higher pressure at greater depths than the modelled water table depth. The model calculates one water level in this case, i.e., the unconfined groundwater level. In the case of artesian water (i.e., a water table elevation higher than the ground surface) the NWT discharges all water above the ground surface. NWT water table

elevation can thus not show artesian water table elevation as the water table is set at the ground surface. Finally, the calibration module of K (see 'Methodology') was shown to yield more realistic groundwater table elevations and K values in the highly-conductive alluvial gravels of the Canterbury Region. It was chosen not to further refine this coarse calibration (e.g., by testing K calibration in the log-domain; or by calibrating more often than only once in 10 years). That is mainly because, in such cases, more advanced local groundwater flow models, including better calibration options, might best be used. These models





could then still be constrained with initial estimates of NWT water tables, hydraulic conductivities, and recharge in data-sparse areas.

Water table depths showed lower correlation to ground observations than water table elevation in the Canterbury and Waipa examples. That is mainly because small inaccuracies of water table depth are much more significant than water table elevations at shallow water tables. For example, in an area that lies 100 metre above sea level, the water table depth error could be highly significant (e.g., $2 \pm 1$ mBGL), where the difference in water table elevation is much less significant (e.g., $98 \pm 1$ masl). The most likely candidate for the cause of such inaccuracies is the uncertainty of the terrain model. For example, Rodríguez et al. (2006) concluded that the SRTM data, on which the national terrain model is partly based, can have average absolute height errors of more than 10 m. Comparison of the LiDAR terrain model in the Waipa evaluation with the national terrain model showed a median absolute difference of 9 m (for areas below 20 masl, the difference was 4 m). Correlation of water table elevation data is independent of the inaccuracy of the terrain models used. However, correlation plots of water table depth proved useful in this study, as they monitored the improvements of model runs by fine-tuning input parametrisation. For example, in the Waipa River catchment correlation of water table depth with ground observations improved from 0.25 to 0.41 when the LiDAR terrain model was used. In both the Canterbury Region and Waipa River catchment we used water table depth correlation to test improvement of the calibration for K.

This research confirms that high-quality model input data, such as hydraulic properties and accurate terrain models are shown to highly important to the improvement of groundwater models. In our opinion, the quality of model input data is more important than the sophistication of a groundwater flow model, since most groundwater flow models are sooner or later calibrated to the locally available (model input) data. More accurate input data than used in the NWT model exist in New Zealand. For example, high resolution mapping of near-surface geology was achieved at the regional scale (Westerhoff et al., 2014) through AEM airborne geophysics and some regional terrain models are based on LiDAR data (e.g., Waikato Regional Council, 2016). However, data acquisition and management of those geophysical and terrain models are at the regional scale. There is no comprehensive national-scale AEM and LiDAR based information as of yet. This study therefore recommends to continue efforts of characterisation of these model input data at the national scale.

Improvements of the NWT model can lead to improved insights of the global-scale model approach. For example, the EWT model crucially requires geology data to infer better water table estimates. The approach in this study, i.e., using hydraulic permeability based on a method of Gleeson et al. (2011), was also used by de Graaf et al. (2015) for the global scale. We recommend that the EWT model approach also embeds these improved input data, and merges other national-scale geological data from other countries. Furthermore, calibration of hydraulic conductivity at locations where water tables are well known, such as done by the NWT model, is a further recommendation that can also improve the EWT model.



## 6 Conclusions

The NWT model is an application of the global-scale EWT model that has been improved for the catchment scale. The NWT model is of finer spatial detail than the EWT model. The NWT model uses adjusted model parametrisation and input datasets, amongst which are a national terrain model and a national digital geological dataset.

The NWT model of New Zealand gives an estimate of water table depth and water table elevation with a 200 m grid resolution. Because of the improvements, NWT water table depths show the areas of alluvial aquifers, including their water table depth variations and their groundwater discharge to surface, better and with higher spatial resolution than the EWT model. The NWT water table elevation shows excellent correlation with ground-observed water level data in the Canterbury Region and Waipa River catchment. The NWT model estimates the water table at places where there are no ground observations, and

shows more detail than other interpolated water table surfaces.

Because of its simplified character, the NWT model also has the advantage of fast calculation at the national scale. In fact, it is currently the only nationwide groundwater model existing in New Zealand. The NWT water table, as well as its nationwide data components (e.g., hydraulic conductivity and recharge), can therefore be used as an initial estimate for more advanced catchment-scale numerical groundwater flow models where data are sparse. In addition, the NWT model might also be used to

solve inconsistency of different regional models at regional boundaries.

Use of the NWT model parametrisation improvements could lead to the improvement of the global-scale EWT model, for example in a better estimation method of hydraulic conductivity. We therefore recommend the findings of this study to be embedded in the EWT model.

The NWT model does not handle complex groundwater features well, i.e., confining layers, fractures, groundwater age,

because the model contains simplifying assumptions. We recommend that state-of-the-art numerical catchment-scale groundwater flow models should be used in those circumstances, if these are available. However, the NWT model is still useful to provide initial model estimates (i.e., water table, hydraulic conductivity and recharge) to those more advanced models.

Possible improvements of the NWT model are the use of better model input components, such as a better terrain model, or improved calibration of hydraulic conductivity. Therefore, this study recommends further efforts in making available high-

quality nationwide geophysical and terrain data at the national scale of New Zealand.





*Acknowledgements.* This research is part of a PhD study of the first author at the University of Waikato, New Zealand, supervised by Prof. Moira Steyn-Ross. It has been performed as part of the Smart Aquifer Characterisation (SAC) Programme, funded by the Ministry of Business, Innovation and Employment, New Zealand. This project has received co-funding from the European Union's Seventh Programme for research technological development and demonstration under grant agreement No 603608, eartH2Observe.



*Competing interests.* There are no competing interests.

*Disclaimer.* TEXT





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



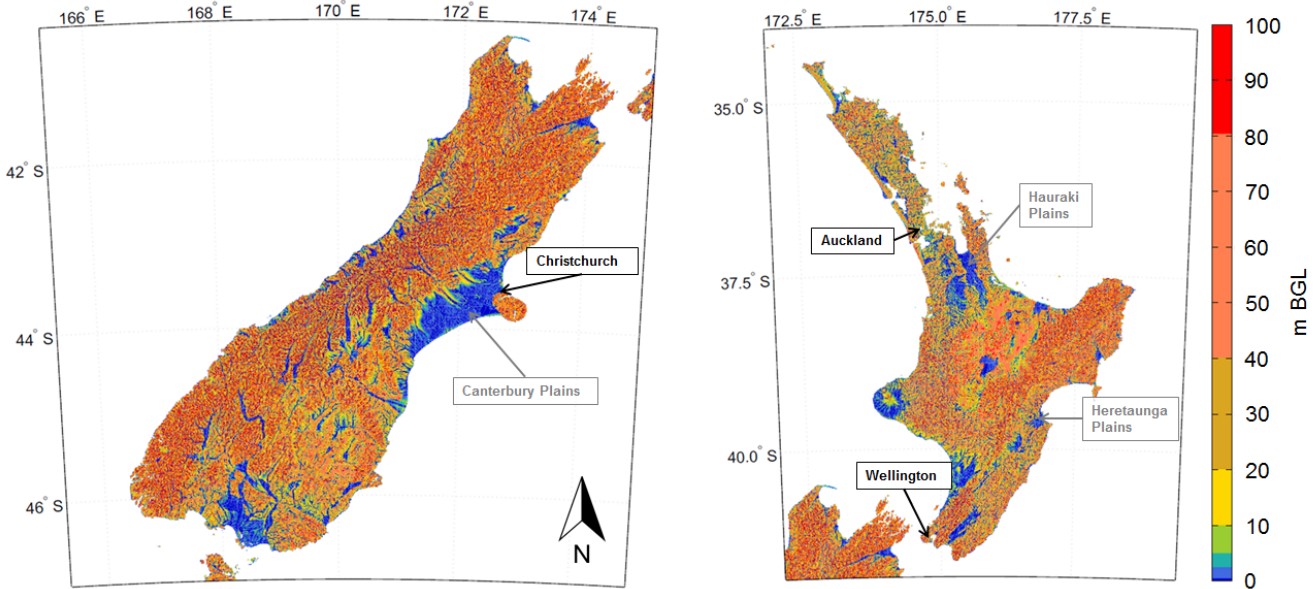

**Figure 1.** EWT water table depth in New Zealand, after Fan et al. (2013a). Main cities (black text boxes) as well as main alluvial aquifer are pointed out (grey text boxes).







**Figure 2.** Hydraulic conductivity estimates for near-surface geology across New Zealand.



**Figure 3.** Mean annual rainfall recharge in New Zealand (2000-2014, Westerhoff et al., 2018).


**Figure 4.** NWT water table depth in New Zealand.







**Figure 5.** NWT water table elevation in New Zealand.



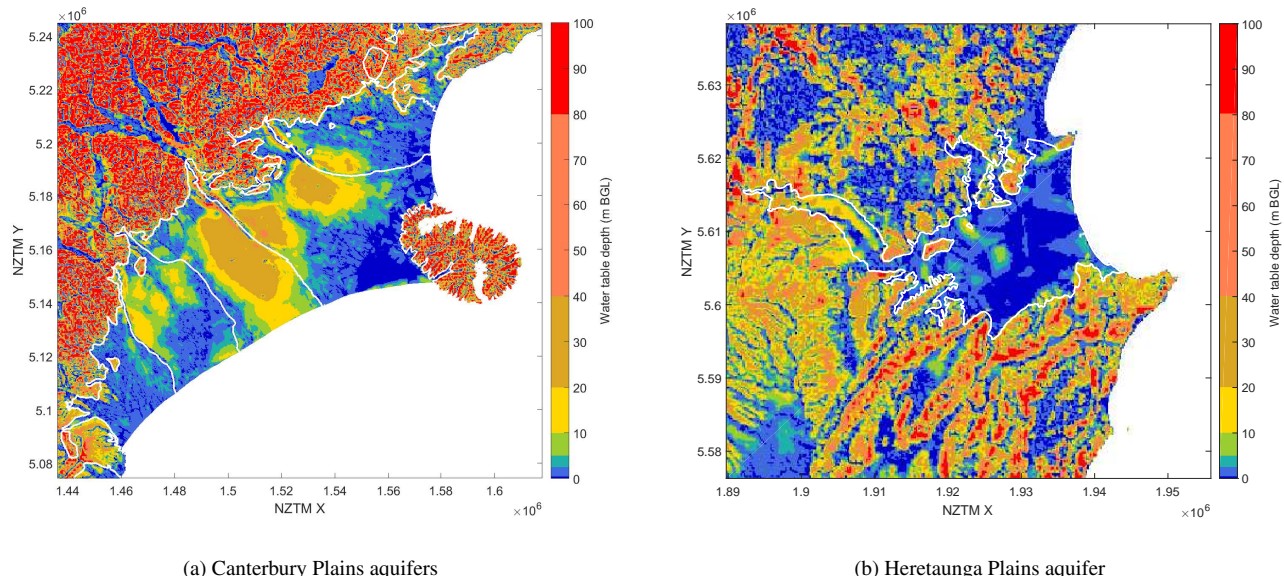

(a) Canterbury Plains aquifers         (b) Heretaunga Plains aquifer

**Figure 6.** NWT water table depth and aquifer boundaries of White (2001) (white lines).

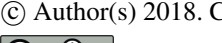




**Figure 7.** Hydrogeological setting of the Canterbury Region (adapted from Brown, 2001).




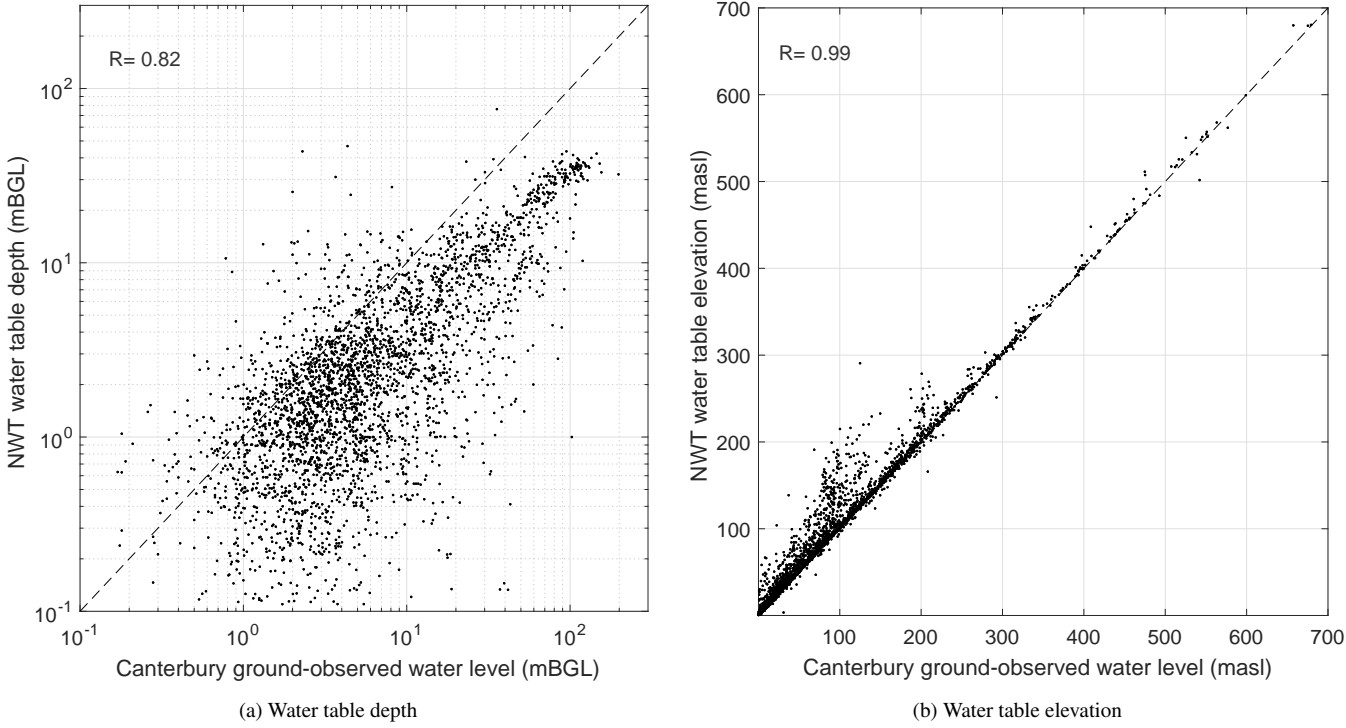

(a) Water table depth

(b) Water table elevation

**Figure 8.** Correlation of NWT water table depth and water table elevation with ground observations in the Canterbury Region. The dashed line is the 1:1 relation.




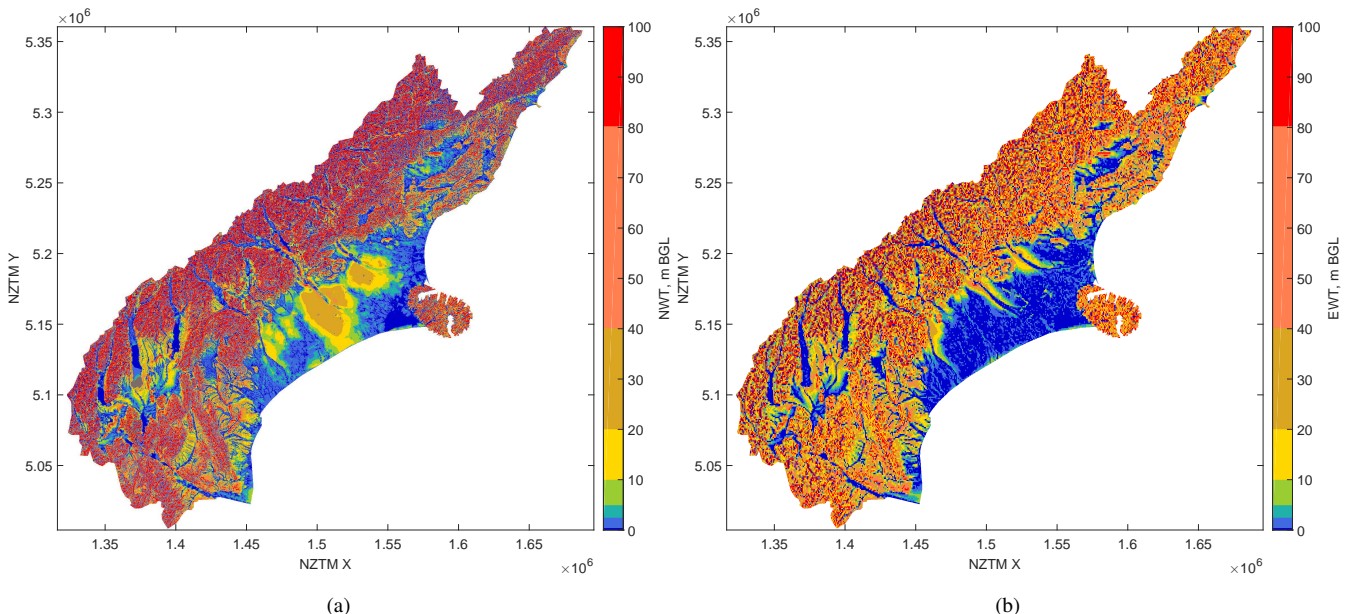

(a)  (b)

**Figure 9.** Water table depth of the (a) NWT and (b) EWT models in the Canterbury Region.






**Figure 10.** Waipa River catchment, Waikato Region, New Zealand. The study area is shown in red. The low-lying Hamilton basin is shown in black (Rawlinson, 2014, their Fig. 1.2).





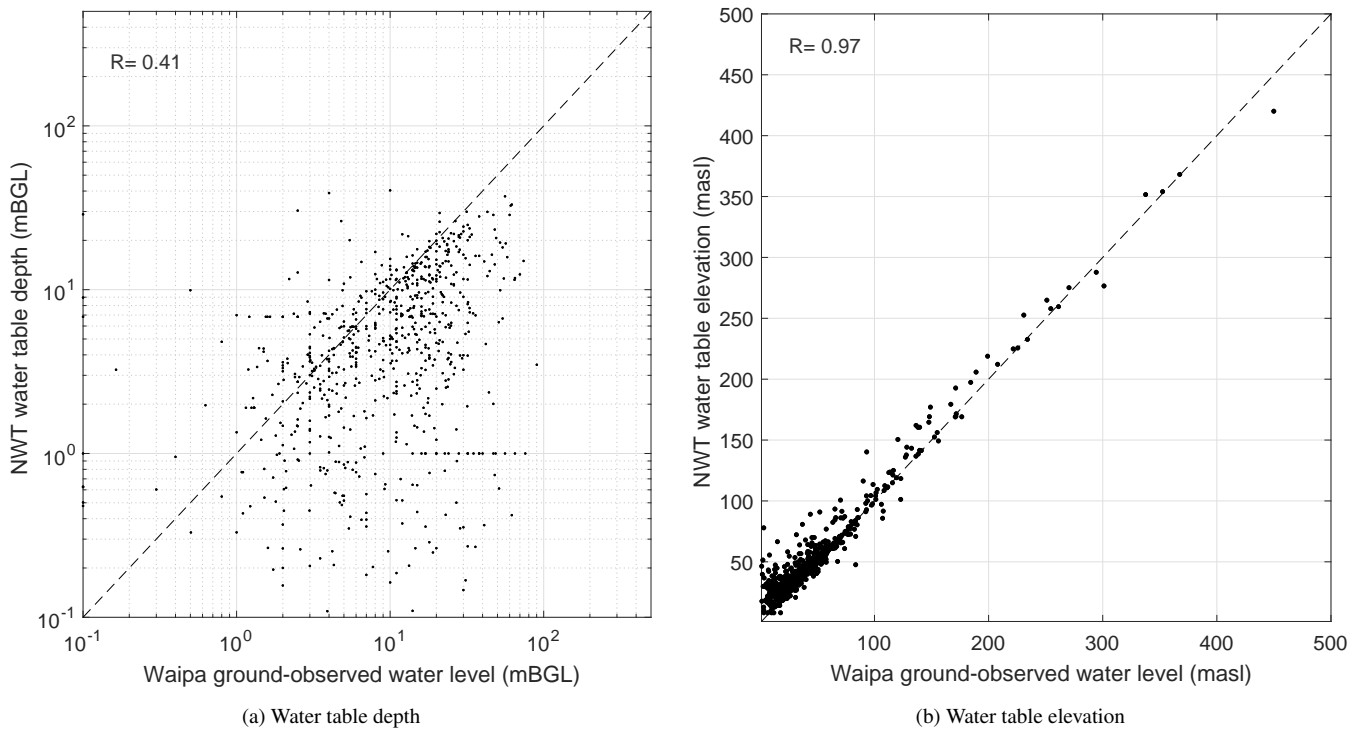

(a) Water table depth

(b) Water table elevation

**Figure 11.** Correlation of NWT water table depth and water table elevation with ground observations in the Waipa River catchment. The dashed line is the 1:1 relation.

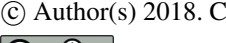



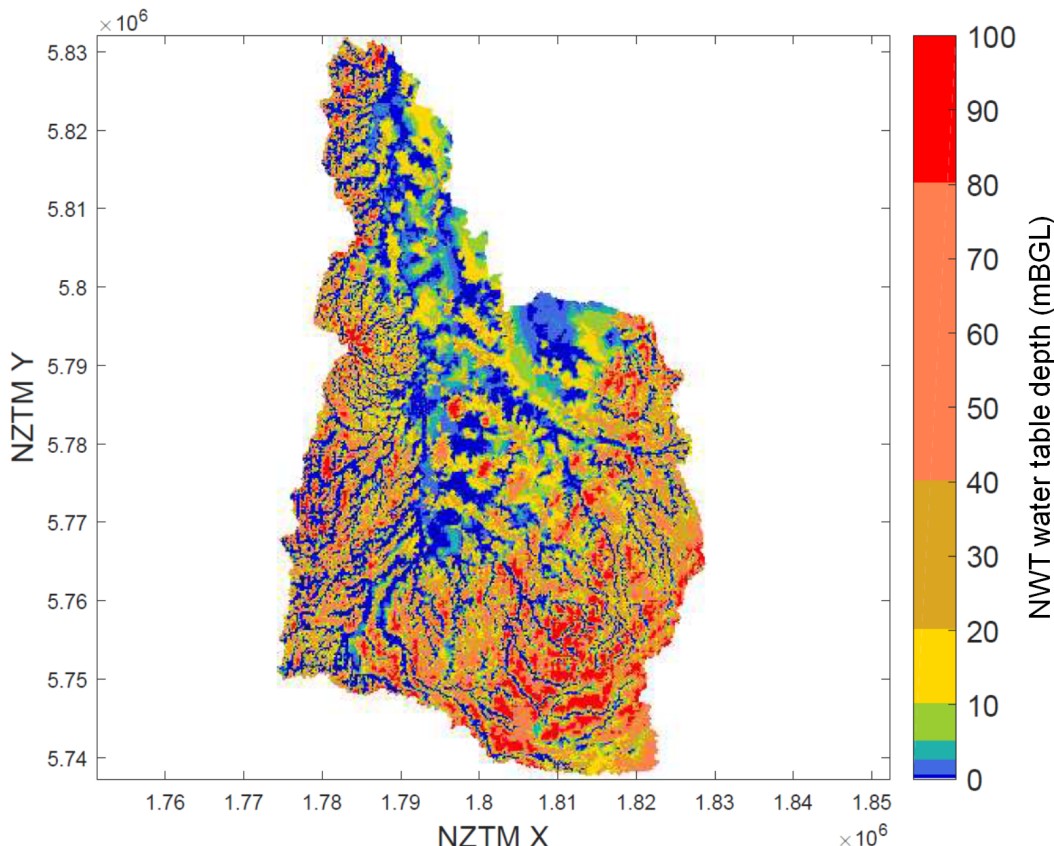

**Figure 12.** NWT water table depth in the Waipa River catchment.





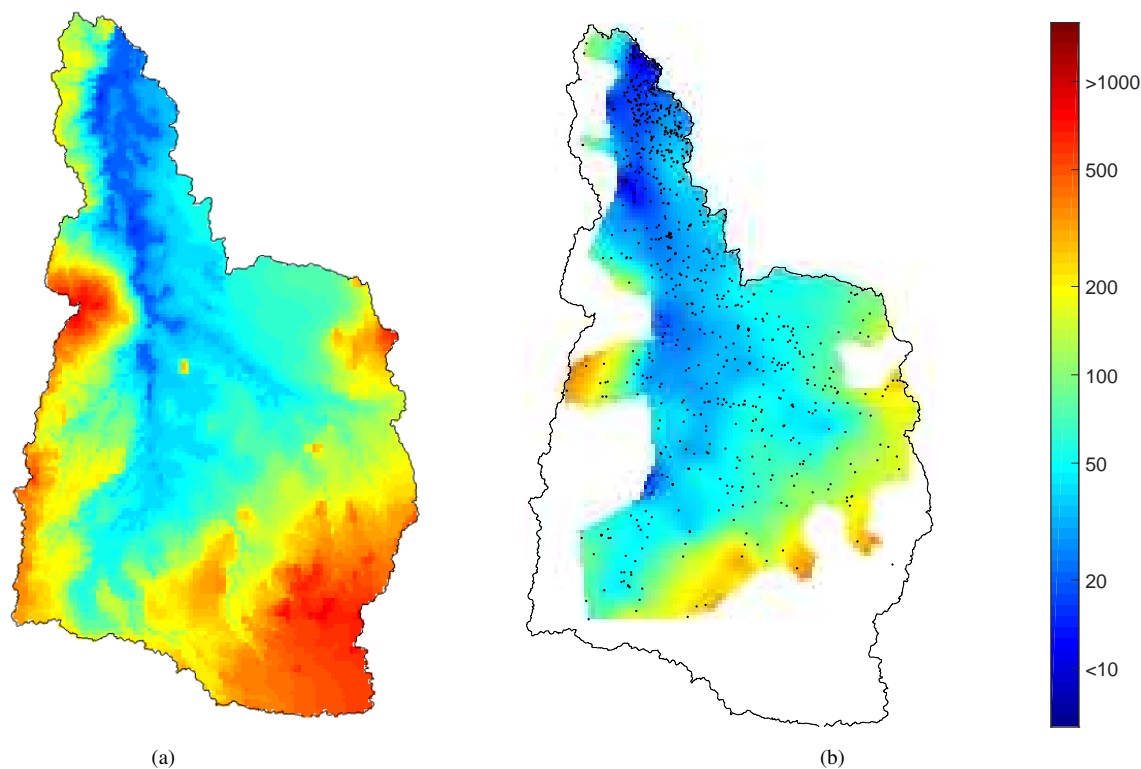

(a)                                                                    (b)

**Figure 13.** (a) NWT water table elevation and (b) the interpolated surface of water table elevation from ground observation in the Waipa catchment (Rawlinson, 2014). Ground observations used for the interpolated surface are shown in black dots.


**Table 1.** New Zealand hydrolithologies and their intrinsic permeability $\kappa$, including standard deviation $\sigma_\kappa$, after Gleeson et al. (2011) and Tschritter et al. (2016). F-g. = fine-grained; p-s. = poorly sorted; c-g. = coarse-grained; uncons. = unconsolidated; sed. = sediment.

| Hydrolithology unit | $\kappa$ [log m$^2$] | $\sigma_\kappa$ [log m$^2$] | Example lithologies |
|---|---|---|---|
| F-g. sed. | -16.5 | 1.7 | mudstone, claystone |
| Crystalline | -15 | 1.5 | granite, greywacke |
| F-g. uncons. sed. | -14 | 1.8 | clay, silt |
| Carbonate | -14 | 1.8 | limestone, shell beds |
| Volcanic | -12.5 | 1.8 | andesite, basalt |
| P-s. sed. | -12.5 | 1.8 | turbidite, breccia |
| P-s. uncons. sed. | -12.5 | 1.8 | peat, till |
| C-g. sed. | -12.5 | 0.9 | sandstone, greenstone |
| Volcanic, high permeability | -11.6 | 1.8 | ignimbrite; scoria |
| C-g. uncons. sed. | -10.5 | 1.2 | gravel; sand |



**Table 2.** Absolute differences between ground-observed and modelled water table depths ($\Delta$) and their percentage of occurrence in the Canterbury Region for the NWT and EWT model.

| $\Delta$ | NWT | EWT |
|---|---|---|
| | (%) | (%) |
| < 1 m | 24 | 13 |
| < 3 m | 53 | 42 |
| > 50 m | 3.4 | 6.0 |
| > 100 m | 0.18 | 1.2 |
| > 150 m | 0.03 | 0.05 |



## Appendix A:  Description of the EWT model

This section summarises the model description of the global-scale EWT model, as described in Fan et al. (2013b).

The EWT model calculates water table depth and water table elevation for a mesh of cells that each have the following properties:

5       - cell size in the horizontal (x,y) directions;

- elevation of the ground surface above sea level;

- annual vertical groundwater recharge from rainfall;

- transmissivity, embedded in a hydraulic conductivity-depth relation;

- annual horizontal groundwater inflow and outflow, which is calculated by the EWT model;

10       - groundwater head, which is calculated by the EWT model.

The cell size for the EWT model is 30 arc-seconds of decimal degrees of latitude and longitude in the WGS84 projection. Therefore, cell size in metres depends on location and varies from 0.76 km east-west and 0.93 km north-south in the North, to 0.63 km east-west and 0.93 km north-south in the South.

The EWT model uses elevation data from global topography models (Smith and Sandwell, 2003; Buis, 2011), all at the 30 arc-second latitude-longitude decimal degree resolution. Rainfall recharge to groundwater is at the 0.5 arc-degree latitude-longitude decimal degree resolution (Döll and Fiedler, 2008). The cell transmissivity is the hydraulic conductivity integrated over depth. Hydraulic conductivity (K) between the ground surface and 1.5 m depth, $K_0$, is derived from a global soil database (Reynolds et al., 2000). Below 1.5 m, K is assumed to decrease exponentially with depth, after the decrease of porosity with depth for large-scale basin studies (Beven and Kirkby, 1979). The exponential decrease of K at depth z below 1.5 m depth is defined as:

$$K(z) = K_0 \, e^{-z/f} \tag{A1}$$

where z is the depth below ground level, and $f$ is called the 'e-folding depth':

$$f = \frac{a}{1+bs}; f > f_{min} \tag{A2}$$

where $a$, $b$, and $f_{min}$ are calibration constants, and $s$ is the terrain slope (Fan et al., 2013b). The inverse relationship of $f$ with $s$ (Eq. A2) is a function of climate, geology and biota (Ahnert, 1970; Summerfield and Hulton, 1994). This relationship causes a large gradient in $K$ over depth where terrain is steeper (i.e., a thin regolith) and a small gradient in K where terrain is relatively flat (i.e., a deep soil). The values of $a$, $b$, and $f_{min}$ are set to 120, 150, and 5, respectively, based on experience of





calibration of the model with ground-observed data in North America (Fan et al., 2013b). No ground-observed water level data from New Zealand have been used to validate Eq. A2.

Groundwater recharge (R) estimates are provided by global-scale mean annual estimates of rainfall recharge to groundwater (Döll and Fiedler, 2008). The horizontal flow between cells (Q) is calculated by Equation A3, based on a mass balance and

Darcy's law (e.g. Hendriks, 2010; Dingman, 2002; Freeze and Cherry, 1979):

$$Q = wT \left( \frac{h - h_n}{L} \right) \tag{A3}$$

where

- $w$ is the width of the cell;

- $T$ is the transmissivity of the cell;

- $h$ is the groundwater head in the centre of the cell;

- $h_n$ is the groundwater head in the neighbouring cell;

- $L$ is the distance between the two cells.

Groundwater discharge into rivers and wetlands ($Q_r$) is identified where groundwater head is above the ground level (Fig. A1). Where the groundwater head rises above the land surface, it is reset to the land surface to mimic the effect of surface

drainage and evaporation. Two important assumptions made in the model are that there is only one water table at any location (thus neglecting local, perched, or multi-layer aquifers) and that groundwater use (i.e., abstraction) is zero.

The EWT model is a steady-state model, where calculations are done iteratively with daily timesteps where recharge is fed into the groundwater flow equation. The calculation converges until an equilibrium between recharge and groundwater flow has been reached, i.e., the mean recharge in a cell equals the mean groundwater flow out of the cell:

$$\overline{R} = \sum \overline{Q} \tag{A4}$$

Computationally, this means that Eq. A3 is repeated until a convergence has been reached, i.e., that the difference in groundwater head between iterations is less than 1 mm in all land cells.

## Appendix B: Convergence tests

Convergence tests were run for the Mataura catchment in Southland, New Zealand (Fig. A2). Tests were measured with: visual

comparison of water table depth; measure of recharge that was rejected due to a shallow water table; and with a convergence ratio, which was defined of the ratio of cells that have a change in hydraulic head of more than 1 cm, measured after each year (365 timesteps). The model was fed with recharge from Westerhoff et al. (2018) in four ways:



1) mean annual recharge was fed in as mean daily recharge;

2) the mean annual recharge was distributed over the year in daily time steps using a normal distribution, i.e., a Gaussian distribution with 365 time steps (Eq. B1):

$$f = \frac{1}{\sigma\sqrt{2\pi}} e^{-\frac{(x-\mu)^2}{2\sigma^2}} \qquad \text{(B1)}$$

where:

- $\sigma$ is the standard deviation of a normal distribution, i.e., 1/6 of the 365 days in this case;

- $\mu$ is the mean, in this case 182.5;

- $x$ is the day in the year, i.e. a vector with values in between 1 and 365.

Eq. B1 mimics a seasonal distribution of rainfall recharge over the year, i.e., seasonal variation of rainfall recharge. Since the model is steady-state, there should not be any difference in model output between seasonal and mean annual recharge input, but the main reason was to test if the NWT model would show a seasonal variation due to different groundwater discharge to surface, as suggested by, e.g., Arnold et al. (2000).

3) as 1), but with incorporation of recharge uncertainty, as estimated by Westerhoff et al. (2018);

4) as 2), but with incorporation of recharge uncertainty, as estimated by Westerhoff et al. (2018);

Inclusion of uncertainty diminishes convergence and given the added noise on the convergence ratio, probably convergence will never be reached. Inclusion of seasonality does not make a difference in convergence when uncertainty is included (Fig. A3) (but converges slightly stronger when uncertainty is not included). None of the four tests show significant differences in water table depths (Fig. A4).

As rejected recharge is increasing with model run time (Table A1), but no visual differences can be seen in the water table depth (Fig. A5), this leads to the conclusion that while the model keeps converging, most of the changes are taking place in the areas with a shallow water table. In reality, most of these areas are drained by humans, which is not taken into account by the model. Therefore, running the model for a too long time to improve model convergence does not significantly improve the water table depth estimates. Because of these findings, for the purpose of estimation of water table depth we chose an efficient 100 year model run time.





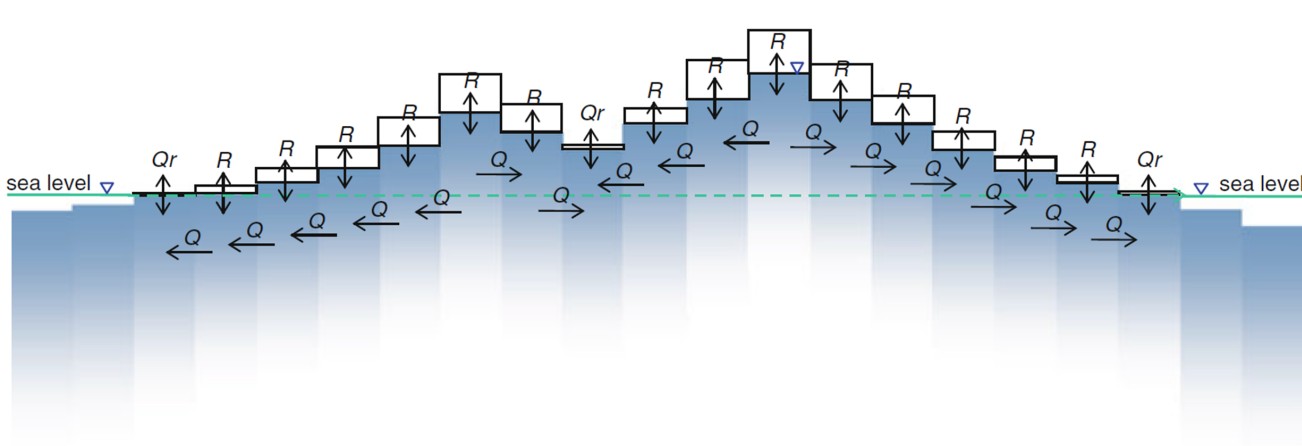

**Figure A1.** Schematic of the EWT model to simulate the water table at the continental scale, using recharge ($R$), horizontal groundwater flow ($Q$), groundwater discharge in rivers ($Q_r$) and sea level (boundary condition). The blue fading colours indicate the decrease of hydraulic conductivity with depth (Fan and Miguez-Macho, 2010b, their Fig. 4a).



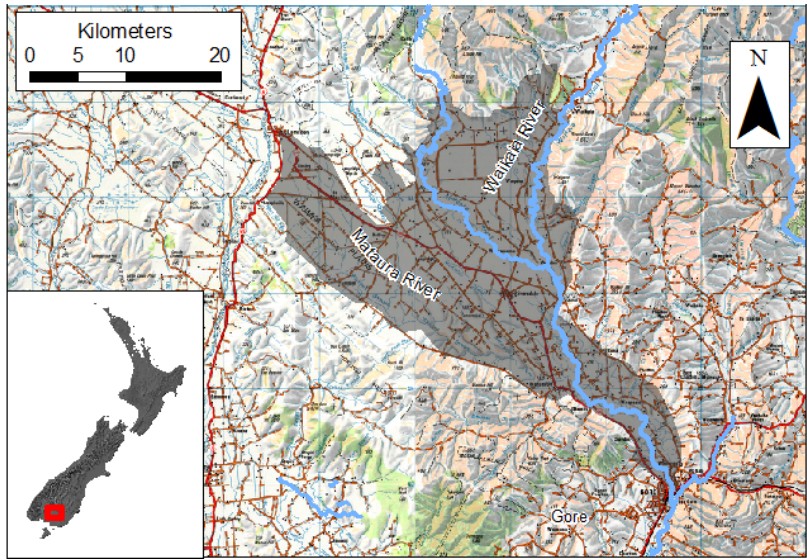

**Figure A2.** Model area of the Mataura catchment, Southland, New Zealand.





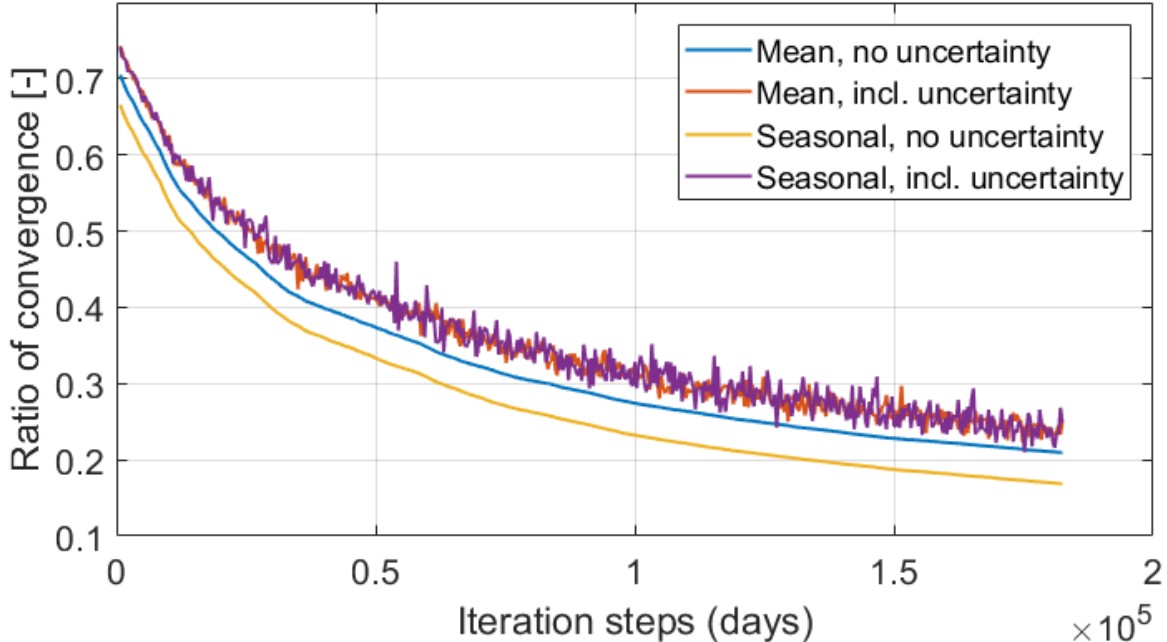

**Figure A3.** Convergence test in the Mataura catchment, Southland, New Zealand, with inclusion of uncertainty and a Gaussian distribution over the year to include seasonality of recharge.





**Figure A4.** Convergence test in the Mataura catchment, Southland, New Zealand, with inclusion of uncertainty and a Gaussian distribution over the year to include seasonality of recharge.





**Figure A5.** Convergence test in the Mataura catchment, Southland, New Zealand, with model run times of 50, 100, 200 and 500 years.





**Table A1.** Volumes of rejected recharge over in the Mataura catchment, Southland, New Zealand, for different model run times.

| model runtime (years) | rejected recharge ($m^3$/s) |
|---|---|
| 50 | 0.66 |
| 100 | 0.84 |
| 200 | 1.03 |
| 500 | 1.31 |