# Peer review of "Application of an improved global-scale groundwater model for water table estimation across New Zealand."

_Hydrology and Earth System Sciences, 2018_

## Referee Comment (RC1) · Anonymous Referee #1 · 13 Jun 2018

The manuscript describes the adaptation of the global steady-state groundwater flow model of Fan et al. (2013), with a spatial resolution of 30 arc-sec (1 km) to New Zealand, resulting in an estimate of depth to groundwater and elevation of the groundwater table everywhere in New Zealand at a spatial resolution of 200 m. Adaptation was mainly done by using input data that were developed specifically for New Zealand with a higher spatial resolution than the input data of the Fan et al. model (Digital elevation model, groundwater recharge) but also using other (likely better) estimates of hydraulic conductivity. They compared model results for two regions with groundwater well data within New Zealand and found a somewhat better fit to the observations than that is achieved with the global scale model.

[Figure]

I suggest rejecting the manuscript for a number of reasons.

A The presentation of the research is done poorly. A1) There are linguistic weakness (e.g. in the abstract: inconsistent terms and meanings with respect to smaller-scale/small-scale/local models and larger-scale/global models, l9: "because the quality of their, coarse and global-scale, input data is large", l1: "larger, i.e. global", while larger here should also refer to national).

A2) The scientific terminology is not always used correctly, and some statements seem to be wrong (e.g. p6, l4 "ground-based, satellite-observed and modelled parameters"; p8 l16: g is not the gravitational constant but gravitational acceleration, and it is not 9.90 $m2/s$ but 9.81 $m/s2$. Why "rainfall recharge" instead of "diffuse groundwater recharge"?

A3) The reason for setting parameter values are not always clearly given, e.g. p8 l27: "As cell resolution of the NWT model is 200 m, the values of a, b and fmin of Eq. A2 were changed accordingly, to 75, 150 and 4". To what extent does the cell size leads to is e.g. setting the value to 75 instead of 120 in the global-scale model?

A4) The manuscript does not explain, except in the Appendix, one major simplification of both the global (EWT) and the New Zealand (NWT) model: There is no hydraulic-gradient dependent interaction between groundwater and rivers; where the water table reaches the land surface, the groundwater is assumed to flow out. This, however, may be the main reason for the dominant overestimation of groundwater table elevation as compared to observations as river levels may be below the land surface elevation. Also, losing rivers cannot be simulated.

B Methodological weaknesses, combined with confusing presentation: In the manuscript, it is stated at various locations that the model was run in daily time steps for 100 years. However, a steady-state model cannot be run at daily time steps, because per definition there is no time variable in a steady state model, and the change in hydraulic head over time is zero. In addition, to do transient runs, one would need to set a storage coefficient, which is not mentioned in the manuscript.

[Figure]

C Lack of new information/innovation that is of general scientific interest The analysis is lacking components that would lead to improved scientific understanding. I suggest to analyze the specific reasons for the better fit of the NWT model results to observations. In a type of sensitivity analysis, variants of the NWT model could be run, in which only one "improved" data set is included while the other data sets remain those of the EWT model. Or alternatively where all but one data set is improved. For example, to understand the impact of the new hydraulic conductivity approach, do one NWT variant in which the approach used in the EWT model is applied. This would be a useful analysis to support the suggestion in the last paragraph of the discussion to use the hydraulic conductivity approach used for NWT (Gleeson data) also for the global-scale EWT, to improve it. But it was not shown in the study whether with the EWT hydraulic conductivity approach the fit to observations in New Zealand would have been better. Similarly, the groundwater recharge estimate used for EWT could be used as input of another NWT variant, and the resulting water table elevations could be compared to the standard NWT results to understand the importance of improved/national groundwater recharge estimates. Then, the presumably large role of the DEM in improving results would be clearer, and your concluding statements would be more firmly based.

I would also suggest adding to Fig. 8 the simulation results of EWT to directly visualize the improvement of NWT over EWT, and adding to Fig. 11 also the results of the standard NWT with a spatial resolution of 200 m, not only the LiDAR-based 100 m variant that is shown (but not indicated in caption).

---

## Author Comment (AC1) · 12 Jul 2018

We thank the referee for the comments and appreciate the effort spent on going so thoroughly through our manuscript.

The reviewer suggests rejecting the manuscript.

We disagree with this recommendation because, in our assessment, minor modifications to the paper will address referee's comments. In our opinion, an objective decision by the editor would see the paper accepted, with minor revisions.

We will stress, in our review of the paper, that the results of our study can be used for national overviews; or to look at inter-catchment (i.e., trans-boundary, across regions) water issues; or as initial estimates for more advanced models. In the light of the special issue from the EartH2Observe project (https://www.hydrol-earth-syst-sci.net/special_issue935.html), this manuscript addresses the collaboration gap between global-scale modellers and catchment-scale modellers by describing a simple method, originating from a global-scale method, that is computationally effective and results in (New Zealand's first) nationwide results.

We have addressed all comments below (*referees comments in Italic* and our reply in red):

*A The presentation of the research is done poorly.*

All of the reviewer's comments are addressed in the following. We largely agree with the comments and will revise the paper accordingly.

*A1) There are linguistic weakness*

*(e.g. in the abstract: inconsistent terms and meanings with respect to smaller scale/*

*small-scale/local models and larger-scale/global models, l9: "because the quality*

*of their, coarse and global-scale, input data is large", l1: "larger, i.e. global", while larger*

*here should also refer to national).*

We agree with the reviewer's comments and will revise the Abstract so that it provides and easier read.

Specifically:

L9: Instead of 'quality', we should have said 'uncertainty', which solves this sentence.

L1: this sentence refers to global-scale models and is therefore correct.

*A2) The scientific terminology is not always used correctly, and some statements seem to be wrong (e.g. p6, l4 "ground-based, satellite-observed and modelled parameters"; p8 l16: g is not the gravitational constant but gravitational acceleration, and it is not 9.90 m2/s but 9.81 m/s2. Why "rainfall recharge" instead of "diffuse groundwater recharge"?*

In regards of P6, l4: "ground-based, satellite-observed and modelled parameters". We can see how this sentence is a little confusing to reader. Minor corrections to grammar will make this sentence easier to understand.

p8 l16: "gravitational acceleration". Yes, that is definitely a typo. Our mistake and we'll correct it. We're also not sure how the 9.9 came in. The value in NZ is approximately 9.80, but we'll change it to the standardised value of 9.81.

*A3) The reason for setting parameter values are not always clearly given, e.g. p8 l27: "As cell resolution of the NWT model is 200 m, the values of a, b and fmin of Eq. A2 were changed accordingly, to 75, 150 and 4". To what extent does the cell size leads to is e.g. setting the value to 75 instead of 120 in the global-scale model?*

We see how p8, l27 – l29 could confuse the reader.  Minor corrections to grammar will make these sentences easier to understand.

In regards of: *'To what extent does the cell size leads to is e.g. setting the value to 75 instead of 120 in the global-scale model?'* We assume that the referee requires explanation as to why these values were used, which is given in the next sentence, that says: "These values were also used by a 200 m resolution EWT model in the Amazon basin (Fan and Miguez-Macho, 2010a) .".

*A4) The manuscript does not explain, except in the Appendix, one major simplification of both the global (EWT) and the New Zealand (NWT) model: There is no hydraulic gradient dependent interaction between groundwater and rivers; where the water table reaches the land surface, the groundwater is assumed to flow out. This, however, may be the main reason for the dominant overestimation of groundwater table elevation as compared to observations as river levels may be below the land surface elevation. Also, losing rivers cannot be simulated.*

Indeed, there is no hydraulic gradient dependent interaction between depending on river water level. More advanced local models, for example modflow (when using one of the integrated  sw processes in modflow) is implicitly coupled at the time step level.  That is, modflow will continue iteratively solving the coupled gw and sw flow equations until the exchange budget error is less than the users specified tolerance.  But the rigor of incorporating a surface water flow model comes at a high cost of model run time and processing cost. We suggest to adjust some texts (minor revision) so that it clarifies better that the novelty of this paper does not lie in looking for the best model. We use an existing global-scale model (in our case EWT) and, knowing that this model is simplified, we are looking at the interplay between model purpose and the simplifications. As mentioned, when looking at simplified, global-scale, and local, advanced, models, we are looking for a model that 'meets in the middle'.

We will add a comment mentioned this limitation in the revised paper. To take away some possible confusion, we could only refer to earlier EWT research references without any further explanation. However, we think that highlighting some model descriptions and simplifications, with a general description in the main text, and slightly more explanation in the Appendix, facilitates an easier read.

This paragraph is only intended for further discussion in case the referee would appreciate that. The underlying assumption in the EWT model is that rivers are resolved in the grid used for integration, which is why calculations have to be done at high resolution. Of course, this is a simplistic approximation, and more so in the original global 30" grid; however, the intent of the global model was to capture continental or global scale (first order) patterns in water table depth, and for this purpose the approach was adequate and results fared surprisingly well when comparing with observations (more than one million data points were used in the global result validation). In the present study, the scope is a much smaller area (New Zealand) and hence grid spacing is much reduced. We are fully aware that at about 200m resolution rivers are still not resolved. Notwithstanding, even when the water table depth might be underestimated in the model simulated

river cells, results in general would not be substantially changed, unless the actual river in the cell runs in a deep and very narrow canyon. The model result in this latter case would represent conditions at the bottom of the canyon, where the river is, which might not be representative of water table depths in the rest of the cell. At 200m resolution, this case would be rather infrequent.

This paragraph is also only intended for further discussion in case the referee would appreciate that. In regard of: *'Also, losing rivers cannot be simulated':* Losing rivers are only simulated by the model assuming that this is recharge. Losing rivers due to river runoff (of water coming from upstream) is indeed not simulated, similar to many numerical groundwater models who also have trouble with this and to this day there is no comprehensive nationwide data of losing and gaining rivers in New Zealand. As mentioned, the EWT and NWT are simplified models, so we do not expect them to pick up model expertise that is not even contained in advanced models. Please note that if losing rivers would be incorporated in the NWT, then the groundwater level would be even higher

*B Methodological weaknesses, combined with confusing presentation: In the manuscript, it is stated at various locations that the model was run in daily time steps for 100 years. However, a steady-state model cannot be run at daily time steps, because per definition there is no time variable in a steady state model, and the change in hydraulic head over time is zero. In addition, to do transient runs, one would need to set a storage coefficient, which is not mentioned in the manuscript.*

The text, which we will correct, is confusing in regards of a steady-state that appears to be transient. In earlier versions of this manuscript we have discussed this with other experts in the field. From that perspective we have chosen to use their advice and call it "steady-state, or dynamic steady-state" since that is the common terminology for such models. So from a physics perspective it is definitely a steady state model. If needed, we can adjust that to 'dynamic steady state' if that satisfies the referee.

We were referring with this comment to the strategy used in achieving convergence in the model result, yielding the sought for equilibrium water table. In the original global calculations with the model, iterations where performed with annual recharge values (i.e. yearly time steps). These speeded up convergence in most parts, however causing fluctuations in high slope terrain with deep water table and substantial recharge, as was the case of the mountains of New Zealand. For this reason, we tried iterations representing smaller time-steps, aiming at limiting the aforementioned fluctuations, and run for a number of steps that we quantified in number of years. Perhaps mentioning years was not a very appropriate choice, since as the reviewer points out, it can cause confusion. We will better clarify in the revised manuscript that the model result is indeed steady-state and not transient, and refer to the number of iterations needed to achieve convergence with the raw number, instead of using a time-measurement equivalent, such as years.

*C Lack of new information/innovation that is of general scientific interest The analysis is lacking components that would lead to improved scientific understanding. I suggest to analyze the specific reasons for the better fit of the NWT model results to observations. In a type of sensitivity analysis, variants of the NWT model could be run, in which only one "improved" data set is included while the other data sets remain those of the EWT model. Or alternatively where all but one data set is improved. For example, to understand the impact of the new hydraulic conductivity approach, do one NWT variant in which the approach used in the EWT model is applied. This would be a useful analysis to support the suggestion in the last paragraph of the discussion to use the hydraulic conductivity approach used for NWT (Gleeson data) also for the global-scale EWT, to improve it. But it was not shown in the study whether with the EWT hydraulic conductivity approach the fit to observations in New Zealand would have been better. Similarly, the groundwater recharge estimate used for EWT could be used as input of another NWT variant, and the resulting water table elevations could be*

*compared to the standard NWT results to understand the importance of improved/national groundwater recharge estimates. Then, the presumably large role of the DEM in improving results would be clearer, and your concluding statements would be more firmly based.*

Re: Lack of new information/innovation that is of general scientific interest.

We contend that the paper is of general scientific interest. The issue of the collaboration gap between global modellers and local, catchment-scale, modellers is a real problem to our science, which focusses at large scale. Interest has been shown in the paper we submitted to the Special Issue "Integration of Earth observations and models for global water resource assessment", forthcoming from the EartH2Observe programme. We also addressed the issue in the EGU 2017 EartH2Observe session called HS1.12 (full session description at http://meetingorganizer.copernicus.org/EGU2017/session/23938). This session was well-attended and is clearly an important future direction for international hydrology.

Re: The analysis is lacking components that would lead to improved scientific understanding.

Improved scientific understanding that comes from the paper, and related work, are:
- that there is a possible role for global-scale models for smaller-scale studies (e.g., national or catchment-scale), i.e., to cover data-sparse areas, to provide initial estimates, to be more computationally efficient.
- That this method covers areas of New Zealand that were never modelled before, and it can thus be used in data-sparse areas as initial estimates without extensive model run time and cost;
- That this model, despite its simplicity, correlates surprisingly well with known observations of hydraulic head. We suggest putting in some graphs on EWT correlation (already quite high) and NWT correlations (even higher correlation), if that satisfies the reviewer.
- The model was used in a way that results in a much more computationally efficient way than any other advanced model. We suggest that we define the model run times of the model for our case study areas and the nation.

To satisfy the reviewer's comments, possible suggestions for improvement to the paper are:

A) To analyse the specific reasons for the better fit of the NWT model results to observations
B) To do one NWT variant in which the approach used in the EWT model is applied.
C) show in the study whether with the EWT hydraulic conductivity approach the fit to observations in New Zealand is better.
D) the groundwater recharge estimate used for EWT could be used as input of another NWT variant, and the resulting water table elevations could be compared to the standard NWT results.

In our opinion, A) and B) would add merit to the paper, without adjusting our intended scope, i.e. point out. Suggestions C) and D) would dive into a deeper uncertainty analysis that we consider outside of the scope of this paper and would typically be undertaken with the incorporation of local advanced models, e.g., to see what improves from EWT to NWT to a local advanced calibrated model (where the 'worth' of a simplified model would be quantified in terms of model purpose, model cost and more). We would typically see this as a (very substantial) follow-up study.

I would also suggest adding to Fig. 8 the simulation results of EWT to directly visualize the improvement of NWT over EWT, and adding to Fig. 11 also the results of the standard NWT with a spatial resolution of 200 m, not only the LiDAR-based 100 m variant that is shown (but not indicated in caption).

Thank you. This is a good idea that we will consider to include in the paper, as an addition to the already described comparisons between EWT and NWT in the current manuscript. results (Section 4.2, Table 2, and Section 4.3).

However, we also want the opinion of the editor if this will not provide too much detail. For example, we could also reference to the PhD thesis in which all these additional figures are depicted to keep the manuscript concise [Westerhoff, R. S. (2017). Satellite remote sensing for improvement of groundwater characterisation (Thesis, Doctor of Philosophy (PhD)). University of Waikato, Hamilton, New Zealand. Retrieved from https://hdl.handle.net/10289/10922)].

---

## Referee Comment (RC2) · Anonymous Referee #2 · 27 Jul 2018

This study presents an updated version of the global-scale equilibrium water table model (Fan et al 2013) for New-Zealand by using New-Zealand specific input data at a higher resolution than done before. This update resulted in an estimate of water table heads and depths at 200m resolution that fit observed water table depths slightly better than before.

I first read the paper to gather my thoughts and then read the comments of reviewer R1, as well as the author response, just to be efficient rather than re-iterating verbosely. I will first say I largely agree with the comments of R1.

I have a few additional comments:

[Figure]

- The writing, and therewith the presentation and discussion of the research, should be significantly be improved. In addition to the points razed by R1, I suggest to rewrite the abstract and introduction and specifically focus on logic of the reasoning (meaning is a statement followed by the right argument and is the argument clear) and being as clear as possible. For example, abstract L2-3 reads: Large-scale models are simplified and not used at smaller-scales, because hydrology and water policy are constrained at the catchment scale. This does not make sense. What the author meant to say is that large scale models, are not useful for smaller scale groundwater assessments yet, because of the simplifications (and the coarse resolutions), therewith are not useful for e.g water policy. The next line reads: However, …... . However, the statement in this line cannot be linked to the previous statement. Something like "for water policy smaller-scale models are more useful. However, …. ." should be included. This are just two examples within the first three lines. Also, be careful using "this" "that" "it" without a summary word.

Overall from the abstract and introduction it was not clear for me what the main motivation and goal of this research were and how it will help us to improve current modelling efforts; to improve the EWT model but also be more useful for water managers? The lack of a logical structure and the bad writing are not beneficial for a clear understanding.

- I found the manuscript very limited in discussion of previous work, methodologies, results, and relevance of the work done.

For example, on discussion of previous work: P4 L4 "many studies …." And then only one reference is a bit limited, as it is not a review paper you refer to. P4 L7 "De Graaf apply a model…. Global-scale input data" This is too generalized, it should be a bit more specific what is meant with "a model" and "input data". Especially as you give some details for the Fan et al 2013 model. 1 to 2 Lines extra focusing on the differences between the two models referred to is needed. I know the models are quit difference. I little review here will also connect to the discussion, and will help you getting your point

across why your model is better than the large-scale models available currently (see also my points later on)

P4 L17: How do you know groundwater models are less reliable in data-sparse regions as there is no data to validate the results. In the case of a model calibration, like done in this study, you can say your model performs best for the regions where you do have data to calibrate on (the whole meaning of a calibration).

Methodology and results: In section 2 it is not explained what happens when water tables hit the surface, nor is it explained that this is not simulated as a head dependent flux and river infiltration (water entering your aquifer) is not included. How realistic is this in the real world? (this should come back in conclusion/discussion as well) Also, your model result look very biased toward shallow water tables, (however not discussed in the manuscript). I think this positive bias can be explained by the way drainage is estimated (see also comment R1).

Another aspect I do not understand is the storage and the convergence criterium that is left out. I agree with R1 that 'steady-state' in combination with a timestep is a bit confusing. How I understand it, is that you run the model over 100 years forced with the same climate data until an equilibrium is reached (i.e. a steady-state). I think for this kind of procedures the term 'dynamic steady state' is used often. (I certainly would not call it transient). What I do not understand, for such a dynamic steady state you still need a storage coefficient, so how does that work? Also, it is not yet clear what you used as a criterion to stop your run. It is written that the convergence criterium is not used, as running the model beyond 100 years did not improve model performance. But how did you decide that 100years were enough; did you check your model outcomes, estimate R for water tables and when that looked good you stopped it. Or was it wall-clock time driven, or CPU time driven? I think whatever criterium you used is fine, but now it raises questions.

I fully agree with R1 on point C and more extensive sensitivity analysis should be done.

From the results it cannot be concluded which model change has the largest impact on the results.

- I agree with R1 and I found the discussion of your work very limited, and not providing enough information to fully understand and acknowledge the importance of the work done.

In my opinion, a relevant aspect of the discussion that is not/not enough elaborated is where we stand now and how it will help is further. How useful is your model in reality, as it is a steady state model approach, not simulating groundwater gradients, calibrated for New-Zealand, under natural conditions only, only unconfined aquifer systems? Are there now model that can do this maybe better, and under real world circumstances (i.e. current climate conditions and human impacts).

In other words, if you need to advice the New-Zealand water managers, how should they use the model and what do they need to know about the model structure and uncertainties to interpret the results correctly and use the model to its full potential? It for which purposes can the model not be used, and what should be improved to make the model useful for the more real world simulation (varying climate and human interactions).

Reading the authors comments on R1 point C I think the authors should be careful in saying that regions where not modelled before (is New Zealand not included in the large-scale models, I think so); stressing the computational efficiency (how efficient is the model, and how does this compare to other large-scale model efforts?).

Minor comments: In the introduction a bit more details on the modelling should be given: 1 to 2 lines saying it is a flux-based approach, simulating steady-state water table heads, using averaged climate conditions, run for 100 years etc.

Related to analysis and scatters: I would suggest to present the R2 (coefficient of determination) is stead of R

P7-L16: "drained by humans"; artificial drainage? P8-L6-7 "who .... ." Leave this out, it is not relevant as you do not use the parameters of Gleeson. P6 L5 "the improved NWT"; is this the same at L4 "the NWT" or is there also an improved version (leave out improved).

F8: it would be more logic to switch those scatters, so that wte, discussed first, becomes (a) and wtd (b) (same for the other scatters).

---

## Author Comment (AC2) · 10 Sep 2018

We thank the referee for the comments and appreciate the effort spent on going so thoroughly through our manuscript. In our opinion, the suggestions from reviewer will improve readability of an improved version a lot.

The reviewer largely agrees with reviewer 1, hence we will not go into too much detail with our general answers, albeit sticking to our points that:

- to address the reviewer's comments would, in our opinion, lead to an an objective decision by the editor to see the paper accepted, with minor revisions.
- we think our manuscript could benefit from a stronger clarification that we are **not** trying to achieve the development of the best possible groundwater model. In the light of this special issue from the EartH2Observe project, this manuscript addresses the collaboration gap between global-scale modellers and catchment-scale modellers by explaining a simple method, originating from a global-scale method, that is computationally effective and results in (New Zealand's first) nationwide results.

We have addressed all comments below (*referees comments in Italic* and our reply in red):

*The writing, and therewith the presentation and discussion of the research, should be significantly be improved. In addition to the points razed by R1, I suggest to rewrite the abstract and introduction and specifically focus on logic of the reasoning (meaning is a statement followed by the right argument and is the argument clear) and being as clear as possible. For example, abstract L2-3 reads: Large-scale models are simplified and not used at smaller-scales, because hydrology and water policy are constrained at the catchment scale. This does not make sense. What the author meant to say is that large scale models, are not useful for smaller scale groundwater assessments yet, because of the simplifications (and the coarse resolutions), therewith are not useful for e.g water policy. The next line reads: However, . . ... . However, the statement in this line cannot be linked to the previous statement. Something like "for water policy smaller-scale models are more useful. However, . . .. ." should be included. This are just two examples within the first three lines. Also, be careful using "this" "that" "it" without a summary word.*

*Overall from the abstract and introduction it was not clear for me what the main motivation and goal of this research were and how it will help us to improve current modelling efforts; to improve the EWT model but also be more useful for water managers? The lack of a logical structure and the bad writing are not beneficial for a clear understanding.*

That is an excellent suggestion, also to clear up the potential confusion arising from the wording about scales. We will incorporate this into our improvements. Thanks.

*- I found the manuscript very limited in discussion of previous work, methodologies, results, and relevance of the work done. For example, on discussion of previous work: P4 L4 "many studies . . .." And then only one reference is a bit limited, as it is not a review paper you refer to. P4 L7 "De Graaf apply a model. . .. Global-scale input data" This is too generalized, it should be a bit more specific what is meant with "a model" and "input data". Especially as you give some details for the Fan et al 2013 model. 1 to 2 Lines extra focusing on the differences between the two models referred to is needed. I know the models are quit difference. I little review here will also connect to the discussion, and will help you getting your point across why your model is better than the large-scale models available currently (see also my points later on)*

Thanks for this suggestion. We will incorporate this into our improved version.

*P4 L17: How do you know groundwater models are less reliable in data-sparse regions as there is no data to validate the results. In the case of a model calibration, like done in this study, you can say your model performs best for the regions where you do have data to calibrate on (the whole meaning of a calibration).*

Instead of the original sentence "Groundwater models are less reliable in data-sparse regions, ......", we will improve this sentence by something like "Because groundwater models cannot be calibrated in data-sparse regions they leads to less reliable results, .....".

*Methodology and results: In section 2 it is not explained what happens when water tables hit the surface, nor is it explained that this is not simulated as a head dependent flux and river infiltration (water entering your aquifer) is not included. How realistic is this in the real world? (this should come back in conclusion/discussion as well) Also, your model result look very biased toward shallow water tables, (however not discussed in the manuscript). I think this positive bias can be explained by the way drainage is estimated (see also comment R1). Another aspect I do not understand is the storage and the convergence criterium that is left out. I agree with R1 that 'steady-state' in combination with a timestep is a bit confusing. How I understand it, is that you run the model over 100 years forced with the same climate data until an equilibrium is reached (i.e. a steady-state). I think for this kind of procedures the term 'dynamic steady state' is used often. (I certainly would not call it transient). What I do not understand, for such a dynamic steady state you still need a storage coefficient, so how does that work? Also, it is not yet clear what you used as a criterion to stop your run. It is written that the convergence criterium is not used, as running the model beyond 100 years did not improve model performance. But how did you decide that 100years were enough; did you check your model outcomes, estimate R for water tables and when that looked good you stopped it. Or was it wallclock time driven, or CPU time driven? I think whatever criterium you used is fine, but now it raises questions.*

*I fully agree with R1 on point C and more extensive sensitivity analysis should be done. From the results it cannot be concluded which model change has the largest impact on the results.*

Since the EWT method has already been explained in great detail in other publications, we feel like it is not our task to explain it again. However, we see the point in having to explain some detail, since we change the method to increase speed. We'll also make sure to include the issue of speed better throughout the manusrcipt: how long would it take to model the whole country with an advanced approach; how fast are we doing it, and what price do we pay in terms of uncertainty. We suggest to rely on an improved uncertainty/sensitivity to explain most of our changes.

*In my opinion, a relevant aspect of the discussion that is not/not enough elaborated is where we stand now and how it will help is further. How useful is your model in reality, as it is a steady state model approach, not simulating groundwater gradients, calibrated for New-Zealand, under natural conditions only, only unconfined aquifer systems? Are there now model that can do this maybe better, and under real world circumstances (i.e. current climate conditions and human impacts). In other words, if you need to advice the New-Zealand water managers, how should they use the model and what do they need to know about the model structure and uncertainties to interpret the results correctly and use the model to its full potential? It for which purposes can the model not be used, and what should be improved to make the model useful for the more real world simulation (varying climate and human interactions).*

We appreciate this comment and will elaborate more on the potential applications where the EWT could be used to solve issues relating to: data-sparsity; national guidelines that cut across regions.

*Reading the authors comments on R1 point C I think the authors should be careful in saying that regions where not modelled before (is New Zealand not included in the large-scale models, I think so); stressing the computational efficiency (how efficient is the model, and how does this compare to other large-scale model efforts?).*
Good point. We will rephrase this.

*Minor comments: In the introduction a bit more details on the modelling should be given: 1 to 2 lines saying it is a flux-based approach, simulating steady-state water table heads, using averaged climate conditions, run for 100 years etc.*

OK. It might even be that we then throw even more of the reimaging theory in the Appendix, depending on how clear our message should be, that we want to use an existing method, with improved data, and see how useful that is to solve water management issues that cut across regions or are in data-sparse areas.

*P7-L16: "drained by humans"; artificial drainage? P8-L6-7 "who . . .. ." Leave this out, it is not relevant as you do not use the parameters of Gleeson. P6 L5 "the improved NWT"; is this the same at L4 "the NWT" or is there also an improved version (leave out improved). F8: it would be more logic to switch those scatters, so that wte, discussed first, becomes (a) and wtd (b) (same for the other scatters).*

Thanks, we appreciate the time taken to even suggest these minor details. We'll improve them.

---

## Author Response (AR1)

We have approved after the consideration of both reviewers and the editors comments. The paper more or less went through a large 're-vamp', where we mostly have addressed the main objections for this paper, which were:

- The pitch and presentation of the paper should be clearer.
  - o We have re-written the introduction so that it should be clear now, that the presentation is: (1) global-scale models are important for global-scale research; (2) advanced local model are important for local studies; (3) but that national-scale modelling is also important and cannot currently be addressed by either global-scale or local-scale models' (4) that we have developed a possible solution that bridges that gap.
- The method description should contain no confusion on steady-state or transient.
  - o We addressed this. See also replies to referee comments below.
- The paper should have more testing results, e.g., sensitivity analyses.
  - o We have put in testing on calibration and different spatial resolutions, but also performed a sensitivity analyses that looks at the sensitivity of the water table to recharge and conductivity.
- The paper should contain more information on why the improved national scale (NWT) model is better than the original global-scale (EWT) model;
  - o We have put in more and clearer comparisons on how the NWT model improves compared to the EWT model in two case studies.
- There should be less confusion about 'scales'.
  - o In the abstract, introduction, discussion and conclusion we are as much as possible sticking to the same presentation as described in the introduction.

We will go over all smaller comments of the reviewers below. We have addressed all comments below (*referees comments in Italic* and our reply in red):

*A1) There are linguistic weakness*

*(e.g. in the abstract: inconsistent terms and meanings with respect to smaller scale/*

*small-scale/local models and larger-scale/global models, l9: "because the quality*

*of their, coarse and global-scale, input data is large", l1: "larger, i.e. global", while larger*

*here should also refer to national).*

Because of the different structure of the introduction, these terms and sentences have been removed altogether. We now tend to use more consistent descriptions of scale.

*A2) The scientific terminology is not always used correctly, and some statements seem to be wrong (e.g. p6, l4 "ground-based, satellite-observed and modelled parameters"; p8 l16: g is not the gravitational constant but gravitational acceleration, and it is not 9.90 m2/s but 9.81 m/s2. Why "rainfall recharge" instead of "diffuse groundwater recharge"?*

We've changed this to "ground-based, satellite-observed and modelled data".

p8 l16: "gravitational acceleration". We've corrected this typo to 9.8 (the approximate value in NZ is 9.80).

*A3) The reason for setting parameter values are not always clearly given, e.g. p8 l27: "As cell resolution of the NWT model is 200 m, the values of a, b and fmin of Eq. A2 were changed accordingly, to 75, 150 and 4". To what extent does the cell size leads to is e.g. setting the value to 75 instead of 120 in the global-scale model?*

We assume that the referee requires explanation as to why these values were used, which is given in the next sentence, that says: "These values were also used by a 200 m resolution EWT model in the Amazon basin (Fan and Miguez-Macho, 2010a) .".

*A4) The manuscript does not explain, except in the Appendix, one major simplification of both the global (EWT) and the New Zealand (NWT) model: There is no hydraulic gradient dependent interaction between groundwater and rivers; where the water table reaches the land surface, the groundwater is assumed to flow out. This, however, may be the main reason for the dominant overestimation of groundwater table elevation as compared to observations as river levels may be below the land surface elevation. Also, losing rivers cannot be simulated.*

According to the earlier discussions on this reviewer comments, we have dedicated a paragraph (p14, l3-18) to this.

*B Methodological weaknesses, combined with confusing presentation: In the manuscript, it is stated at various locations that the model was run in daily time steps for 100 years. However, a steady-state model cannot be run at daily time steps, because per definition there is no time variable in a steady state model, and the change in hydraulic head over time is zero. In addition, to do transient runs, one would need to set a storage coefficient, which is not mentioned in the manuscript.*

We were referring with this comment to the strategy used in achieving convergence in the model result, yielding the sought for equilibrium water table. In the original global calculations with the model, iterations where performed with annual recharge values (i.e. yearly time steps). These speeded up convergence in most parts, however causing fluctuations in high slope terrain with deep water table and substantial recharge, as was the case of the mountains of New Zealand. For this reason, we tried iterations representing smaller time-steps, aiming at limiting the aforementioned fluctuations, and run for a number of steps that we quantified in number of years. Perhaps mentioning years was not a very appropriate choice, since as the reviewer points out, it can cause confusion. We have now better clarified in the revised manuscript that the model result is indeed steady-state and not transient, and refer to the number of iterations needed to achieve convergence with the raw number, instead of using a time-measurement equivalent, such as years.

*C Lack of new information/innovation that is of general scientific interest The analysis is lacking components that would lead to improved scientific understanding. I suggest to analyze the specific reasons for the better fit of the NWT model results to observations. In a type of sensitivity analysis, variants of the NWT model could be run, in which only one "improved" data set is included while the other data sets remain those of the EWT model. Or alternatively where all but one data set is improved. For example, to understand the impact of the new hydraulic conductivity approach, do one NWT variant in which the approach used in the EWT model is applied. This would be a useful analysis to support the suggestion in the last paragraph of the discussion to use the hydraulic conductivity approach used for NWT (Gleeson data) also for the global-scale EWT, to improve it. But it was not*

*shown in the study whether with the EWT hydraulic conductivity approach the fit to observations in New Zealand would have been better. Similarly, the groundwater recharge estimate used for EWT could be used as input of another NWT variant, and the resulting water table elevations could be compared to the standard NWT results to understand the importance of improved/national groundwater recharge estimates. Then, the presumably large role of the DEM in improving results would be clearer, and your concluding statements would be more firmly based.*

As we submitted to the Special Issue "Integration of Earth observations and models for global water resource assessment" forthcoming from the EartH2Observe programme, we addressed the issue that was also addressed in the EGU 2017 EartH2Observe session called HS1.12 (full session description at http://meetingorganizer.copernicus.org/EGU2017/session/23938), i.e., the collaboration gap between global modellers and local, catchment-scale, modellers.

The improved scientific understanding in our manuscript is that there is a possible role for global-scale models for smaller-scale studies (e.g., national or catchment-scale). After the reviewer's comments we chose to show improvement of NWT to EWT results (Section 4.2 , Table 2, and Section 4.3).

I would also suggest adding to Fig. 8 the simulation results of EWT to directly visualize the improvement of NWT over EWT, and adding to Fig. 11 also the results of the standard NWT with a spatial resolution of 200 m, not only the LiDAR-based 100 m variant that is shown (but not indicated in caption).

We have added this, as well as a substantial addition where model resolution is analysed.

Reviewer 2: specific comments:

*The writing, and therewith the presentation and discussion of the research, should be significantly be improved. In addition to the points razed by R1, I suggest to rewrite the abstract and introduction and specifically focus on logic of the reasoning (meaning is a statement followed by the right argument and is the argument clear) and being as clear as possible. For example, abstract L2-3 reads: Large-scale models are simplified and not used at smaller-scales, because hydrology and water policy are constrained at the catchment scale. This does not make sense. What the author meant to say is that large scale models, are not useful for smaller scale groundwater assessments yet, because of the simplifications (and the coarse resolutions), therewith are not useful for e.g water policy. The next line reads: However, . . ... . However, the statement in this line cannot be linked to the previous statement. Something like "for water policy smaller-scale models are more useful. However, . . ... ." should be included. This are just two examples within the first three lines. Also, be careful using "this" "that" "it" without a summary word.*

*Overall from the abstract and introduction it was not clear for me what the main motivation and goal of this research were and how it will help us to improve current modelling efforts; to improve the EWT model but also be more useful for water managers? The lack of a logical structure and the bad writing are not beneficial for a clear understanding.*

We have restructured the abstract and introduction so that the message comes across better now.

*- I found the manuscript very limited in discussion of previous work, methodologies, results, and relevance of the work done. For example, on discussion of previous work: P4 L4 "many studies . . .." And then only one reference is a bit limited, as it is not a review paper you refer to.*

We have improved it, so all of the mentioned topics now have two references. Also, we use the word 'existing' instead of 'many'.

*P4 L7 "De Graaf apply a model. . .. Global-scale input data" This is too generalized, it should be a bit more specific what is meant with "a model" and "input data". Especially as you give some details for the Fan et al 2013 model. 1 to 2 Lines extra focusing on the differences between the two models referred to is needed. I know the models are quit difference. I little review here will also connect to the discussion, and will help you getting your point across why your model is better than the large-scale models available currently (see also my points later on)*

We have incorporated this into our introduction.

*P4 L17: How do you know groundwater models are less reliable in data-sparse regions as there is no data to validate the results. In the case of a model calibration, like done in this study, you can say your model performs best for the regions where you do have data to calibrate on (the whole meaning of a calibration).*

This text, with its paragraph, has been removed to have an altogether clearer introduction.

*Methodology and results: In section 2 it is not explained what happens when water tables hit the surface, nor is it explained that this is not simulated as a head dependent flux and river infiltration (water entering your aquifer) is not included. How realistic is this in the real world? (this should come back in conclusion/discussion as well) Also, your model result look very biased toward shallow water tables, (however not discussed in the manuscript). I think this positive bias can be explained by the way drainage is estimated (see also comment R1). Another aspect I do not understand is the storage and the convergence criterium that is left out. I agree with R1 that 'steady-state' in combination with a timestep is a bit confusing. How I understand it, is that you run the model over 100 years forced with the same climate data until an equilibrium is reached (i.e. a steady-state). I think for this kind of procedures the term 'dynamic steady state' is used often. (I certainly would not call it transient). What I do not understand, for such a dynamic steady state you still need a storage coefficient, so how does that work? Also, it is not yet clear what you used as a criterion to stop your run. It is written that the convergence criterium is not used, as running the model beyond 100 years did not improve model performance. But how did you decide that 100years were enough; did you check your model outcomes, estimate R for water tables and when that looked good you stopped it. Or was it wallclock time driven, or CPU time driven? I think whatever criterium you used is fine, but now it raises questions.*

*I fully agree with R1 on point C and more extensive sensitivity analysis should be done. From the results it cannot be concluded which model change has the largest impact on the results.*

We have addressed these topics:

In the model review. We have added extra text on the bias to shallow water tables, caused by the fact that the model does not incorporate pumping or drainage. Still, we try to keep the model review short in the main text (as the EWT has already been described in full detail in earlier studies) with further references to an appendix.

Furthermore, we have added text that explains that the NWT model still has a bias towards shallow water tables, although less than the EWT model because of the finer model resolution. However, this bias of shallow water table is also a correct indicator of the fact that most of the indicated

shallow water table areas used to be wetlands: an approximate 90\% of wetlands have been lost since European settlement in New Zealand, mostly to develop agriculture.

We have also addressed this topic in the discussion, where we explain that the model resolves for river better, but still does not incorporate draining features.

We now have a clearer description where we leave out the '100 years' if needed, so it is extremely clear we have a steady-state model.

*In my opinion, a relevant aspect of the discussion that is not/not enough elaborated is where we stand now and how it will help is further. How useful is your model in reality, as it is a steady state model approach, not simulating groundwater gradients, calibrated for New-Zealand, under natural conditions only, only unconfined aquifer systems? Are there now model that can do this maybe better, and under real world circumstances (i.e. current climate conditions and human impacts). In other words, if you need to advice the New-Zealand water managers, how should they use the model and what do they need to know about the model structure and uncertainties to interpret the results correctly and use the model to its full potential? It for which purposes can the model not be used, and what should be improved to make the model useful for the more real world simulation (varying climate and human interactions).*

We have now hopefully clarified better, through a better and clearer introduction, on the potential applications where the EWT could be used to solve issues relating to: data-sparsity; national guidelines that cut across regions.

*Reading the authors comments on R1 point C I think the authors should be careful in saying that regions where not modelled before (is New Zealand not included in the large-scale models, I think so); stressing the computational efficiency (how efficient is the model, and how does this compare to other large-scale model efforts?).*
We have rephrased, also saying that this is the first 'dedicated' national groundwater model for New Zealand.

*Minor comments: In the introduction a bit more details on the modelling should be given: 1 to 2 lines saying it is a flux-based approach, simulating steady-state water table heads, using averaged climate conditions, run for 100 years etc.*

We have added slightly more elaboration in the introduction, but also to the model review, where we think it is more appropriate.

*P7-L16: "drained by humans"; artificial drainage? P8-L6-7 "who . . .. ." Leave this out, it is not relevant as you do not use the parameters of Gleeson. P6 L5 "the improved NWT"; is this the same at L4 "the NWT" or is there also an improved version (leave out improved). F8: it would be more logic to switch those scatters, so that wte, discussed first, becomes (a) and wtd (b) (same for the other scatters).*

Thanks, we've improved those.

---

## Author Response (AR2)

Dear editor,
Thanks for your positive response. We have addressed the last minor comments in the attached pdf with the following explanations:

1) Section 3.3.3 and response to A3: "These values were also used by a 200m resolution EWT model in the Amazon basin (Fan and Miguez-Macho, 2010a)." In the revised version you removed the values of the calibration factors a, b, fmin from the text and refer to the Appendix. But in the Appendix you give numbers for a, b and fmin based on experience of model calibration in North America. Please clarify!

We have changed this wording in the main text as:

"Near-surface K was assumed to be represented by the QMAP-derived near-surface K (Fig. 2). Deeper than 10 m, an exponential decrease of hydraulic conductivity over depth was assumed, similar to Eq. A1. As cell resolution of the NWT model is 200 m,the calibration constants used in this exponential decrease (a,b and fmin, see Appendix, Eq. A2) were set equal to an earlier developed 200 m resolution and locally calibrated EWT model in the Amazon basin (Fan and Miguez-Macho, 2010a)"

And in the Appendix, we have changed the wording to:

"The values of a,b, and fmin for the global 30-arc second EWT model are set to 120, 150, and 5, respectively, based on experience of calibration of the model with ground-observed data in North America (Fan et al.,2013b). No ground-observed water level data from New Zealand have been used to validate Eq. A2. For a case study in the Amazon, with a more detailed 200m resolution model, Fan and Miguez-Macho (2010a) used revised values for a,b, and fmin (75, 150, and 4, respectively)."

2) A reference to Fig. 10 is not set in the text.
Done. We had to put the figures in the right order so the text referred to 8 first, then 9 and then 10.

3)
We have changed the sentence: "Also, this study that improved input of (and calibration towards) hydraulic conductivity further reduces this bias." to "This study shows that improved input data of hydraulic conductivity (or calibration towards it) further reduces this bias."

Finally, our latex document is made in Overleaf. Can we submit through Overleaf, or do we have to send the original tex files? No problems whatsoever for both options, but through Overleaf might be more efficient.

Kind regards,
Rogier Westerhoff